# Non-invasive assessment of telomere maintenance mechanisms in brain tumors

Pavithra Viswanath [1✉], Georgios Batsios [1], Joydeep Mukherjee[2], Anne Marie Gillespie[1], Peder E. Z. Larson [1], H. Artee Luchman[3], Joanna J. Phillips [2], Joseph F. Costello[2], Russell O. Pieper[2] & Sabrina M. Ronen[1✉]

Telomere maintenance is a universal hallmark of cancer. Most tumors including low-grade oligodendrogliomas use telomerase reverse transcriptase (TERT) expression for telomere maintenance while astrocytomas use the alternative lengthening of telomeres (ALT) pathway. Although TERT and ALT are hallmarks of tumor proliferation and attractive therapeutic targets, translational methods of imaging TERT and ALT are lacking. Here we show that TERT and ALT are associated with unique $^1$H-magnetic resonance spectroscopy (MRS)-detectable metabolic signatures in genetically-engineered and patient-derived glioma models and patient biopsies. Importantly, we have leveraged this information to mechanistically validate hyperpolarized [1-$^{13}$C]-alanine flux to pyruvate as an imaging biomarker of ALT status and hyperpolarized [1-$^{13}$C]-alanine flux to lactate as an imaging biomarker of TERT status in low-grade gliomas. Collectively, we have identified metabolic biomarkers of TERT and ALT status that provide a way of integrating critical oncogenic information into non-invasive imaging modalities that can improve tumor diagnosis and treatment response monitoring.

[1] Department of Radiology and Biomedical Imaging, University of California San Francisco, San Francisco, CA, USA. [2] Department of Neurological Surgery, Helen Diller Research Center, University of California San Francisco, San Francisco, CA, USA. [3] Department of Cell Biology and Anatomy, Arnie Charbonneau Cancer Institute and Hotchkiss Brain Institute, University of Calgary, Calgary, AB, Canada. ✉email: pavithra.viswanath@ucsf.edu; sabrina.ronen@ucsf.edu

Telomere maintenance mechanisms (TMMs) are fundamental hallmarks of cancer[1]. Telomeres are repetitive, nucleoprotein structures that protect chromosomal ends from DNA damage[2]. They shorten with every cell division until the cell undergoes either senescence or apoptosis[2]. In order to proliferate indefinitely, tumor cells must acquire a TMM[1–3]. Telomerase reverse transcriptase (TERT) expression is the predominant TMM in cancer, including in low-grade oligodendrogliomas (LGOGs)[2–5]. TERT is the catalytic subunit of the enzyme telomerase that synthesizes telomeric DNA and its expression is silenced at birth in normal somatic cells, with the exception of stem cells[3]. Mutations in the TERT promoter rank among the most common noncoding mutations in cancer[3–5] and lead to the creation of a novel binding site for the transcription factor GABP, which reactivates TERT expression[6,7]. In contrast to LGOGs, low-grade astrocytomas (LGAs) activate a TERT-independent, homologous recombination-mediated mechanism of new telomeric DNA synthesis known as the alternative lengthening of telomeres (ALT) pathway[8–10]. Loss-of-function mutations in the chromatin remodeling protein, alpha-thalassemia/mental retardation syndrome X-linked (ATRX), which occur in the majority of LGAs, are associated with induction of the ALT pathway, and ATRX re-expression has been shown to repress the ALT pathway[4,11–13].

TMMs are attractive therapeutic targets. TERT promoter mutations occur exclusively in tumor cells and not in normal somatic cells or stem cells[3]. Although directly targeting TERT activity results in toxicity due to stem cell inhibition[3], inhibiting TERT expression from the mutant TERT promoter by disrupting GABP function is a viable therapeutic strategy[6]. ALT is also the focus of drug discovery[8,14]. Inhibition of the protein kinase ATR, which is a critical regulator of the ALT pathway, selectively kills ALT-positive gliomas, and the brain-penetrant ATR inhibitors berzosertib and AZD6738 are currently in clinical trials for solid tumors[15].

Previous studies have linked TERT to metabolic reprogramming[16]. Pharmacological or genetic inhibition of TERT abrogates glucose flux through the pentose phosphate pathway[17] and de novo fatty acid synthesis in gliomas[18]. TERT has also been linked to altered redox status and, specifically, to higher levels of reduced glutathione (GSH) and reduced oxidative stress in cancer cells[19]. Although less is understood about the links between ALT and metabolism, studies show that ALT cells downregulate the expression of the telomere-associated protein RAP1[13] which has been linked to metabolism[20].

Clinical glioma patient management is heavily dependent on non-invasive magnetic resonance imaging (MRI) methods such as T1-weighted pre- and post-gadolinium, T2/FLAIR, and diffusion-weighted imaging[21] since the anatomical location of gliomas and their infiltrative nature can prohibit biopsy sampling and/or surgical resection[22,23]. However, anatomic MRI is inadequate for the detection of molecular events driving glioma proliferation and fails to distinguish "true" tumor from morphologically similar areas of gliosis, edema, and necrosis[24,25]. Importantly, there are no reliable methods to distinguish tumor recurrence from treatment-related effects such as pseudoprogression and pseudoresponse[26,27].

A complementary approach is magnetic resonance spectroscopy (MRS), which is a safe, nonradioactive method of imaging metabolism in live cells, animals, and patients[28–30]. Since oncogenic events are often linked to metabolic reprogramming[1,31], MRS can provide a downstream readout of underlying tumor genetics. [1]H-MRS measures steady-state metabolite levels and is used in the clinic[28]. Thermally polarized [13]C-MRS following administration of [13]C-labeled precursors informs on metabolic fluxes, but its low sensitivity limits its translational value[28]. However, the recent development of hyperpolarized [13]C-MRS has

enhanced the signal-to-noise ratio (SNR) by >10,000-fold and provides a translational method of imaging metabolic fluxes[29,30].

As molecular hallmarks of tumor proliferation, TMMs are diagnostic biomarkers of "true" tumor burden. Imaging TMM status, therefore, has the potential to non-invasively differentiate tumor from the normal brain and from anatomically indistinguishable areas of gliosis, edema, necrosis, pseudoprogression, and pseudoresponse. Imaging TMMs can also inform on response to chemoradiotherapy and emerging TMM inhibitors. In this context, the link between TMMs and metabolic reprogramming provides an opportunity to leverage TMM-linked metabolic alterations for non-invasive, MRS-detectable imaging of TMM status in LGOGs and LGAs.

Here we show that, in genetically-engineered and patient-derived LGOG and LGA models as well as in patient biopsies, TERT is linked to elevated [1]H-MRS-detectable levels of NAD(P)/H, GSH, aspartate, and AXP (combined signal from ATP, ADP, and AMP) while the ALT pathway leads to elevated α-ketoglutarate (α-KG), glutamate, alanine, and AXP. Importantly, we have exploited this information to identify and mechanistically validate hyperpolarized [1-[13]C]-alanine metabolism to pyruvate as an in vivo imaging biomarker of the ALT pathway in LGAs and hyperpolarized [1-[13]C]-alanine flux to lactate as an in vivo imaging biomarker of TERT expression in LGOGs.

## Results

### TERT expression and the ALT pathway induce unique [1]H-MRS-detectable metabolic signatures in genetically-engineered glioma models.

We began our studies with genetically-engineered, immortalized normal human astrocyte (NHA) models[32] that express mutant isocitrate dehydrogenase 1 (IDHmut), which is a defining feature of both LGOGs and LGAs[33,34]. We examined cells that were engineered to endogenously reactivate TERT expression (NHA$_{TERT}$) or activate the ALT pathway following ATRX loss (NHA$_{ALT}$)[13,32,35,36]. As controls, we examined cells that expressed only IDHmut and lacked both TERT or the ALT pathway (NHA$_{CONTROL}$)[13,35]. TERT expression and telomerase activity have been confirmed in NHA$_{TERT}$ cells (previously in ref. [35] and Supplementary Fig. 1a, b). Similarly, in line with the previous study[13], we verified that loss of ATRX in NHA$_{ALT}$ cells (Supplementary Fig. 1c, d) resulted in activation of the ALT pathway by confirming the presence of c-circles (Supplementary Fig. 1e), which are extrachromosomal circles of single-stranded telomeric DNA that are considered to be specific and quantifiable markers of the ALT pathway[37,38]. The ALT status of NHA$_{ALT}$ cells has also previously been confirmed via detection of telomeric sister chromatid exchange (T-SCE), which is another phenotypic hallmark of the ALT pathway[13]. Supplementary Table 1 provides a summary of the cell lines and their IDHmut and TMM status.

We examined steady-state metabolite levels in cell extracts using [1]H-MRS (Fig. 1a). In order to identify TMM-linked metabolic alterations in an unbiased manner, we subjected the [1]H-MRS data to principal component analysis (PCA). As shown in Fig. 1b, PCA discriminated the metabolic profiles of NHA$_{CONTROL}$, NHA$_{TERT}$, and NHA$_{ALT}$ cells. Analysis of the variable importance in projection (VIP) scores from partial least squares discriminant analysis (PLSDA)[39] indicated that elevated NAD(P)/H (combined signal from NAD+, NADP+, NADH, and NADPH since these entities cannot be reliably distinguished by [1]H-MRS), GSH, aspartate, and AXP in NHA$_{TERT}$ cells discriminated from NHA$_{CONTROL}$ (Supplementary Fig. 2a). Increased α-KG, glutamate, alanine, and AXP in NHA$_{ALT}$ cells discriminated from NHA$_{CONTROL}$ (Supplementary Fig. 2b). Elevated NAD(P)/H, GSH, and aspartate in NHA$_{TERT}$ and higher α-KG, glutamate, and alanine in NHA$_{ALT}$ discriminated between these two models (Supplementary Fig. 2c). Univariate

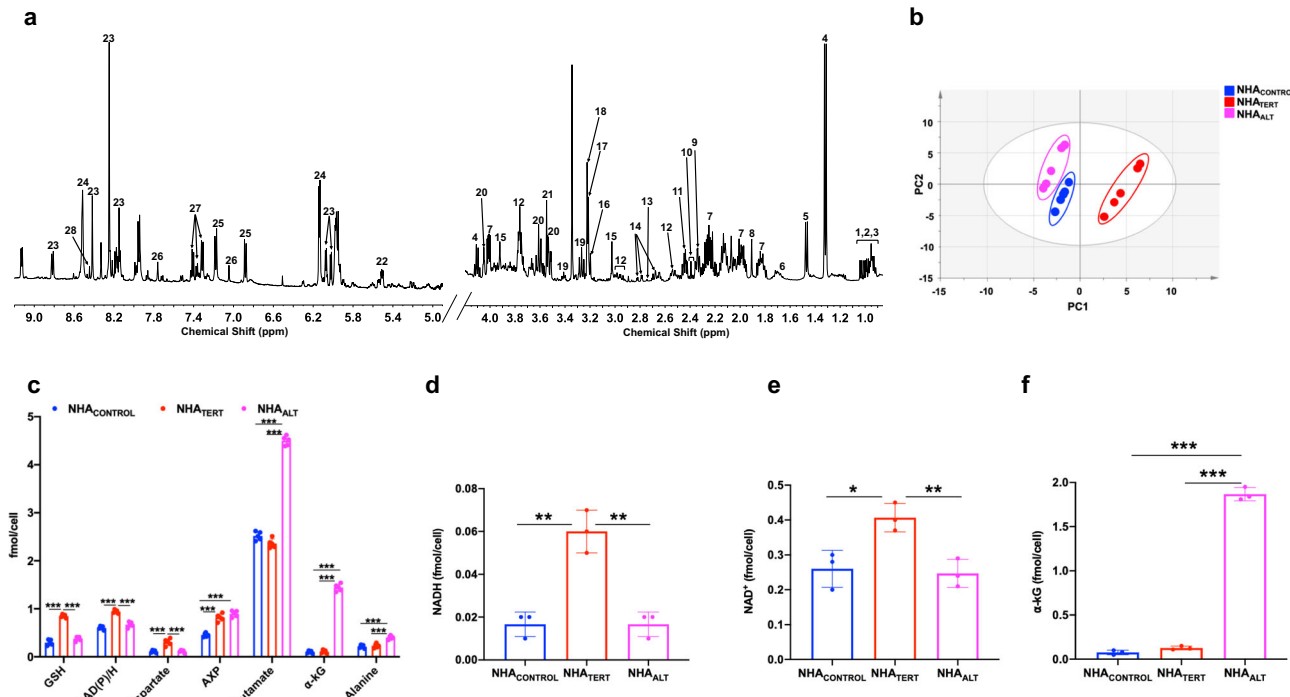

**Fig. 1 TERT expression and the ALT pathway are associated with unique $^1$H-MRS detectable metabolic signatures in low-grade glioma cells.**
**a** Representative $^1$H-MRS spectrum from NHA$_{TERT}$ cells. 1–3: isoleucine, leucine, valine; 4: lactate; 5: alanine; 6: leucine; 7: 2-hydroxyglutarate; 8: acetate; 9: glutamate; 10: α-KG; 11: glutamine; 12: glutathione; 13: cystathionine; 14: aspartate; 15: creatine; 16: choline; 17: phosphocholine; 18: glycerophosphocholine; 19: taurine; 20: myo-inositol: 21: glycine; 22: UDP-N-acetylglucosamine; 23: NAD(P)/H 24: AXP; 25: tyrosine; 26: histidine 27: phenylalanine; 28: formate.
**b** Scores plot of PCA for NHA$_{CONTROL}$ (blue circles), NHA$_{TERT}$ (red circles), and NHA$_{ALT}$ (magenta circles) cells ($n = 5$ biological replicates each).
**c** Metabolite concentrations quantified from $^1$H-MRS spectra for NHA$_{CONTROL}$ (blue circles), NHA$_{TERT}$ (red circles), and NHA$_{ALT}$ (magenta circles) models ($n = 5$ biological replicates each). Levels of NADH (**d**), NAD$^+$ (**e**), and α-KG (**f**) measured by spectrophotometric assays in NHA$_{CONTROL}$ (blue circles), NHA$_{TERT}$ (red circles), and NHA$_{ALT}$ (magenta circles) models ($n = 3$ biological replicates each). Results are presented as mean ± standard deviation. Statistical significance was assessed using an unpaired Student's $t$ test assuming unequal variance with $p < 0.05$ considered significant. Correction for multiple comparisons was performed using the Holm–Šídák method. *** represents statistical significance with $p < 0.005$; ** represents statistical significance with $p < 0.01$; * represents statistical significance with $p < 0.05$. Also, refer to Supplementary Fig. 2 and Supplementary Fig. 3. Source data and exact $p$ values are provided as a source data file.

analysis of absolute metabolite levels quantified from $^1$H-MRS spectra confirmed the statistical significance of these results ($p < 0.005$ for all metabolites; Fig. 1c). We further measured levels of NAD$^+$, NADP$^+$, NADH, NADPH, α-KG, GSH, and oxidized glutathione using spectrophotometric assays. Consistent with previous studies linking TERT to redox status[17–19], our results indicated that NADH (Fig. 1d) and NAD$^+$ (Fig. 1e) were significantly higher in NHA$_{TERT}$ cells relative to NHA$_{CONTROL}$ and NHA$_{ALT}$, leading to a reduced NAD$^+$/NADH ratio (Supplementary Fig. 2d). α-KG was higher in NHA$_{ALT}$ cells relative to NHA$_{CONTROL}$ and NHA$_{TERT}$ (Fig. 1f) while NADPH (Supplementary Fig. 2e) and GSH (Supplementary Fig. 2f) were significantly higher in NHA$_{TERT}$ cells relative to NHA$_{CONTROL}$ and NHA$_{ALT}$. There was no difference in oxidized glutathione (Supplementary Fig. 2g) or NADP$^+$ (Supplementary Fig. 2h) between NHA$_{CONTROL}$, NHA$_{TERT}$, or NHA$_{ALT}$ models.

Our studies identified similar levels of the oncometabolite 2-hydroxyglutarate (2-HG), the product of the IDHmut enzyme[33,34], between NHA$_{CONTROL}$, NHA$_{TERT}$, and NHA$_{ALT}$ models (Supplementary Fig. 3), suggesting that the metabolic biomarkers identified above were linked to TERT or ALT status as opposed to IDHmut. Nevertheless, in order to further assess the individual contributions of *IDHmut*, *TERT*, and *ATRX*, we also examined NHAs that do not express *IDHmut* and are either *ATRX*-deficient (NHA$_{ATRX-IDHmut−}$)[13] or express *TERT* (NHA$_{TERT+IDHmut−}$)[35]. Lack of *IDHmut* expression in these cell lines has previously been

confirmed[13,35]. We also confirmed the absence of 2-HG in NHA$_{ATRX-IDHmut−}$ and NHA$_{TERT+IDHmut−}$ cells via $^1$H-MRS (see Supplementary Fig. 3). Previous studies indicate that *ATRX* loss in the absence of *IDHmut* does not lead to the development of the ALT pathway[13]. In line with these studies, we confirmed the absence of c-circles in NHA$_{ATRX-IDHmut−}$ cells (Supplementary Fig. 1e). Lack of T-SCE has also been previously confirmed in these cells[13]. Importantly, in contrast to NHA$_{ALT}$ cells, $^1$H-MRS-detectable levels of glutamate, α-KG, alanine, or AXP were not altered in NHA$_{ATRX-IDHmut−}$ cells relative to NHA$_{CONTROL}$ cells, thereby linking these metabolites to the ALT pathway (Supplementary Fig. 3). Similarly, in contrast to NHA$_{TERT}$ cells, levels of GSH, NAD(P)/H, aspartate, and AXP were not altered in NHA$_{TERT+IDHmut−}$ cells relative to NHA$_{CONTROL}$ cells (see Supplementary Fig. 3). Collectively, the results presented in this section point to GSH, NAD(P)/H, aspartate, and AXP as potential $^1$H-MRS-detectable biomarkers of TERT status and to glutamate, α-KG, alanine, and AXP as potential biomarkers of ALT status in low-grade gliomas harboring IDHmut.

**Hyperpolarized [1-$^{13}$C]-alanine can non-invasively monitor TMM status in genetically-engineered glioma models.** α-KG participates in the conversion of alanine to pyruvate mediated by the enzyme alanine aminotransferase (see schematic in Fig. 2a). Further conversion of pyruvate to lactate catalyzed by lactate

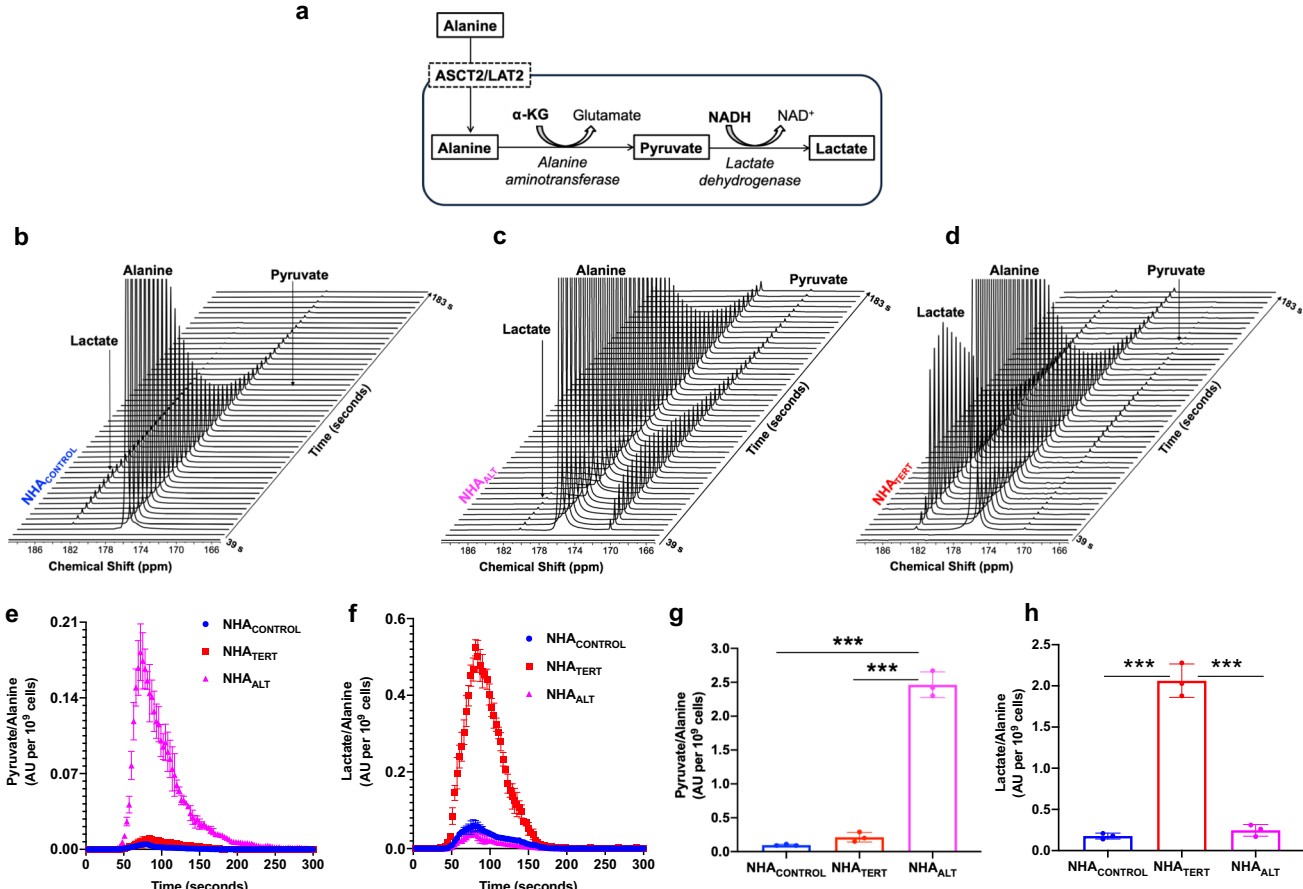

**Fig. 2 Hyperpolarized [1-¹³C]-alanine can non-invasively monitor TMM status in low-grade glioma cells. a** Schematic illustration of alanine metabolism to pyruvate and lactate. Hyperpolarized [1-¹³C]-alanine is imported into the cell via the alanine transporters ASCT2 or LAT2. Subsequent metabolism of alanine to pyruvate is dependent on the concomitant conversion of α-KG to glutamate in a reaction mediated by alanine aminotransferase. Further reduction of pyruvate to lactate by lactate dehydrogenase involves the simultaneous conversion of NADH to NAD⁺. Representative ¹³C spectral array showing hyperpolarized [1-¹³C]-alanine metabolism in live NHA_CONTROL (**b**), NHA_ALT (**c**), and NHA_TERT (**d**) cells. Spectral arrays are not scaled to cell number. Dynamic build-up curves for hyperpolarized [1-¹³C]-pyruvate (**e**) and hyperpolarized [1-¹³C]-lactate (**f**) following injection of hyperpolarized [1-¹³C]-alanine in NHA_CONTROL (blue circles), NHA_TERT (red squares), and NHA_ALT (magenta triangles) cells. Pyruvate/alanine (**g**) and lactate/alanine (**h**) ratios for NHA_CONTROL (blue circles), NHA_TERT (red circles), and NHA_ALT (magenta circles) cells calculated from the summed ¹³C-MRS spectra from the spectral arrays. All experiments were performed on three biological replicates (*n* = 3). Results are presented as mean ± standard deviation. Statistical significance was assessed using an unpaired Student's *t* test assuming unequal variance with *p* < 0.05 considered significant. *** represents statistical significance with *p* < 0.005. Source data with exact *p* values are provided as a source data file.

dehydrogenase is dependent on NADH[40,41]. Since our data indicated that α-KG was elevated in NHA_ALT cells while NADH was elevated in NHA_TERT cells (see Fig. 1c, d), we examined the ability of hyperpolarized [1-¹³C]-alanine to probe TMM status. [1-¹³C]-alanine was polarized (T1 = 44 ± 4 s, 12% polarization, consistent with previous studies[42,43]), added to a suspension of live cells and dynamic ¹³C spectra acquired. NHA_CONTROL cells showed minimal production of [1-¹³C]-pyruvate and [1-¹³C]-lactate, consistent with limiting levels of α-KG and NADH in these cells (Fig. 2b). In contrast, NHA_ALT cells produced higher levels of [1-¹³C]-pyruvate, consistent with elevated α-KG (Fig. 2c). However, further conversion to [1-¹³C]-lactate was minimal due to limiting NADH. NHA_TERT cells, which have higher NADH, converted hyperpolarized [1-¹³C]-alanine to [1-¹³C]-pyruvate and then onto [1-¹³C]-lactate (Fig. 2d). These results are highlighted in the build-up curves showing maximal production of [1-¹³C]-pyruvate in NHA_ALT cells (Fig. 2e) and maximal production of [1-¹³C]-lactate in NHA_TERT (Fig. 2f). Quantification indicated a significantly higher pyruvate/alanine ratio in NHA_ALT cells relative to NHA_TERT and NHA_CONTROL

(Fig. 2g) and a significantly higher lactate/alanine ratio in NHA_TERT cells relative to NHA_ALT and NHA_CONTROL (Fig. 2h). Collectively, these results suggest that differential metabolism of hyperpolarized [1-¹³C]-alanine to either [1-¹³C]-pyruvate or [1-¹³C]-lactate non-invasively monitors TMM status in low-grade glioma cells.

**¹H-MRS and hyperpolarized [1-¹³C]-alanine can track modulation of TERT expression or the ALT pathway in patient-derived LGOG and LGA models**. Next, we examined whether modulation of TERT expression and the ALT pathway in clinically relevant, patient-derived LGOG and LGA models resulted in modulation of their corresponding ¹H-MRS-detectable metabolic alterations and hyperpolarized [1-¹³C]-alanine metabolism. To this end, for TERT status, we investigated BT54 neurospheres which were originally derived from a patient carrying an *IDHmut* LGOG tumor[44–46], and examined the effect of silencing *TERT* using two nonoverlapping siRNA oligonucleotides directed against *TERT*. We confirmed a significant reduction in *TERT*

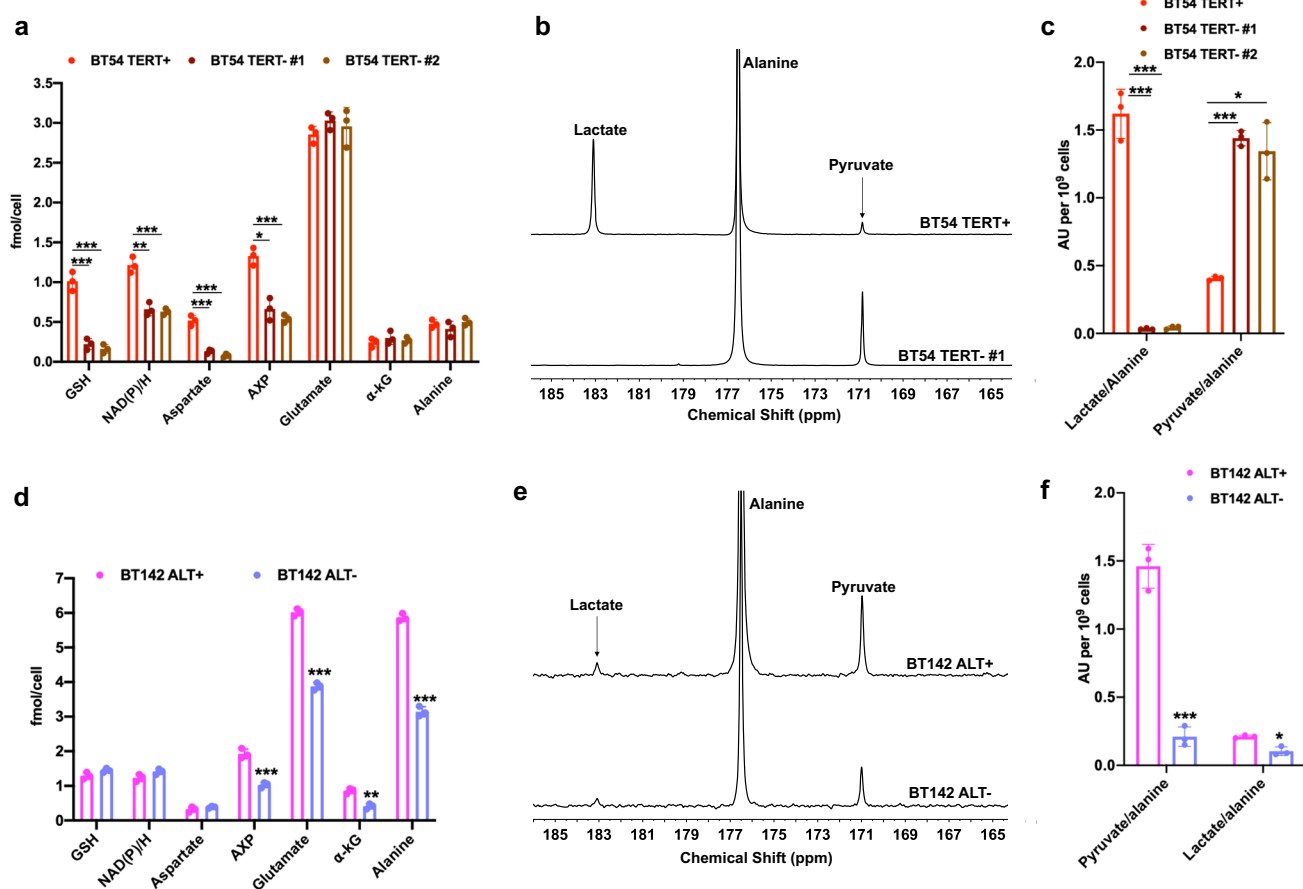

**Fig. 3 Silencing TERT expression or the ALT pathway normalizes $^1$H-MRS biomarkers and hyperpolarized [1-$^{13}$C]-alanine metabolism in patient-derived LGOG and LGA models. a** Effect of *TERT* silencing on steady-state metabolite levels as measured by $^1$H-MRS in the BT54 LGOG model (BT54 TERT + neurospheres: red circles; BT54 TERT− #1 neurospheres: dark brown circles; BT54 TERT− #2 neurospheres: light brown circles). *TERT* silencing was carried out with two independent, nonoverlapping siRNA sequences/pools. **b** Representative summed $^{13}$C-MRS spectra showing hyperpolarized [1-$^{13}$C]-alanine metabolism in BT54 TERT+ and BT54 TERT− neurospheres. **c** Quantification of the ratios for hyperpolarized lactate/alanine and hyperpolarized pyruvate/alanine in BT54 TERT+ and TERT− neurospheres (BT54 TERT+ neurospheres: red circles; BT54 TERT− #1 neurospheres: dark brown circles; BT54 TERT− #2 neurospheres: light brown circles). *TERT* silencing was carried out with two independent, nonoverlapping siRNA sequences/pools. **d** Effect of silencing the ALT pathway via *ATRX* re-expression on steady-state metabolite levels as measured by $^1$H-MRS in the BT142 LGA model (BT142 ALT+ neurospheres: magenta circles; BT142 ALT− neurospheres (lavender circles). **e** Representative summed $^{13}$C-MRS spectra showing hyperpolarized [1-$^{13}$C]-alanine metabolism in BT142 ALT+ and BT142 ALT− neurospheres. **f** Quantification of the ratios for hyperpolarized pyruvate/alanine and hyperpolarized lactate/alanine in BT142 ALT+ (magenta circles) and BT142 ALT− (lavender circles) neurospheres. All experiments were performed on three biological replicates ($n = 3$). Results are presented as mean ± standard deviation. Statistical significance was assessed using an unpaired Student's $t$ test assuming unequal variance with $p < 0.05$ considered significant. Correction for multiple comparisons was performed using the Holm–Šídák method. *** represents statistical significance with $p < 0.005$; ** represents statistical significance with $p < 0.01$; * represents statistical significance with $p < 0.05$. Also, refer to Supplementary Fig. 4. Source data with exact $p$ values are provided as a source data file.

expression as well as telomerase activity (Supplementary Fig. 1a, b), without accompanying effects on cell proliferation (Supplementary Fig. 4a). As shown in Fig. 3a, $^1$H-MRS-detectable levels of metabolites linked to TERT in the NHA$_{TERT}$ model (refer to Fig. 1c), i.e. NAD(P)/H, GSH, aspartate, and AXP were significantly reduced in BT54 TERT− neurospheres relative to BT54 TERT+. Spectrophotometric assays confirmed that both NADH and NAD$^+$ were reduced in BT54 TERT− neurospheres relative to BT54 TERT+, leading to a higher NAD$^+$/NADH ratio (Supplementary Fig. 4b–d). In contrast, levels of metabolites linked to ALT status, i.e., α-KG, glutamate, and alanine were not altered in BT54 TERT− neurospheres (Fig. 3a). Importantly, consistent with reduced NADH following TERT silencing, hyperpolarized [1-$^{13}$C]-alanine was metabolized to [1-$^{13}$C]-pyruvate but further metabolism to [1-$^{13}$C]-lactate was abrogated in BT54 TERT− cells (Fig. 3b), leading to a significant drop in the hyperpolarized

lactate/alanine ratio and a corresponding increase in the hyperpolarized pyruvate/alanine ratio (Fig. 3c).

For interrogation of ALT status, we examined BT142 neurospheres which were derived from an *ATRX*-deficient IDHmut LGA tumor[47] and have been shown to use the ALT pathway for telomere maintenance[13]. In line with previous studies[12,13], we suppressed the ALT pathway in BT142 neurospheres by transient re-expression of *ATRX* (see Supplementary Fig. 1c for corroboration of ATRX mRNA levels and Supplementary Fig. 4e for ATRX protein levels). We established that the ALT pathway was inhibited in BT142 ALT− neurospheres by confirming the loss of c-circles (Supplementary Fig. 1e) and T-SCE (Supplementary Fig. 1g). Suppression of the ALT pathway did not inhibit BT142 proliferation (Supplementary Fig. 4f). Consistent with our results in the NHA models, assessment of steady-state metabolite levels by $^1$H-MRS revealed significant reductions in α-KG, glutamate,

alanine, and AXP in BT142 ALT− neurospheres relative to BT142 ALT+ (Fig. 3d). In contrast, levels of metabolites linked to TERT, i.e., NAD(P)/H, GSH, and aspartate were unaltered by inhibition of the ALT pathway (Fig. 3d). Importantly, consistent with reduced α-KG in BT142 ALT− neurospheres, hyperpolarized [1-13C]-alanine metabolism to pyruvate was significantly reduced (Fig. 3e). Quantification of the data showed significant reductions in the hyperpolarized pyruvate/alanine and lactate/alanine ratios in BT142 ALT− neurospheres relative to BT142 ALT+ (Fig. 3f).

In addition to assessing the effect of silencing *TERT* expression or the ALT pathway as described above, we also interrogated the validity of our metabolic imaging biomarkers in isogenic patient-derived models that were engineered to be either ALT+ or TERT+. Previous studies suggest that exogenous *TERT* expression in ALT+ cells can potentially lead to a TERT+ phenotype, thereby providing an isogenic platform to interrogate TMM status[48–50]. We, therefore, examined two pairs of patient-derived LGA models (BT142 ALT+ and MGG119 ALT+) that were engineered to express *TERT* (BT142 TERT+ and MGG119 TERT+). MGG119

neurospheres were isolated from an *ATRX*-deficient IDHmut LGA tumor[51] and have previously been shown to use the ALT pathway as their TMM[13]. Here, we confirmed that MGG119 ALT+ neurospheres use the ALT pathway as evidenced by the loss of *ATRX* expression (Supplementary Fig. 1c), presence of c-circles at levels comparable to those observed in U2OS cells, which are considered a standard for the ALT phenotype[37,38] (Supplementary Fig. 1e) and occurrence of T-SCE (Supplementary Fig. 1f). In contrast, MGG119 TERT+ and BT142 TERT+ neurospheres showed TERT expression and telomerase activity (Supplementary Fig. 1a, b) accompanied by loss of c-circle formation (Supplementary Fig. 1e) and loss of T-SCE (Supplementary Fig. 1f). It is important to note that levels of *TERT* expression and telomerase activity in MGG119 TERT+ and BT142 TERT+ neurospheres were comparable to the endogenous levels observed in the NHA_TERT and patient-derived BT54 LGOG model (see Supplementary Fig. 1a, b). As shown in Fig. 4a, b, 1H-MRS-detectable levels of TERT-linked metabolites (GSH, NAD(P)/H and aspartate) were significantly higher and levels of ALT-linked metabolites (α-KG, glutamate, and alanine) were significantly lower in

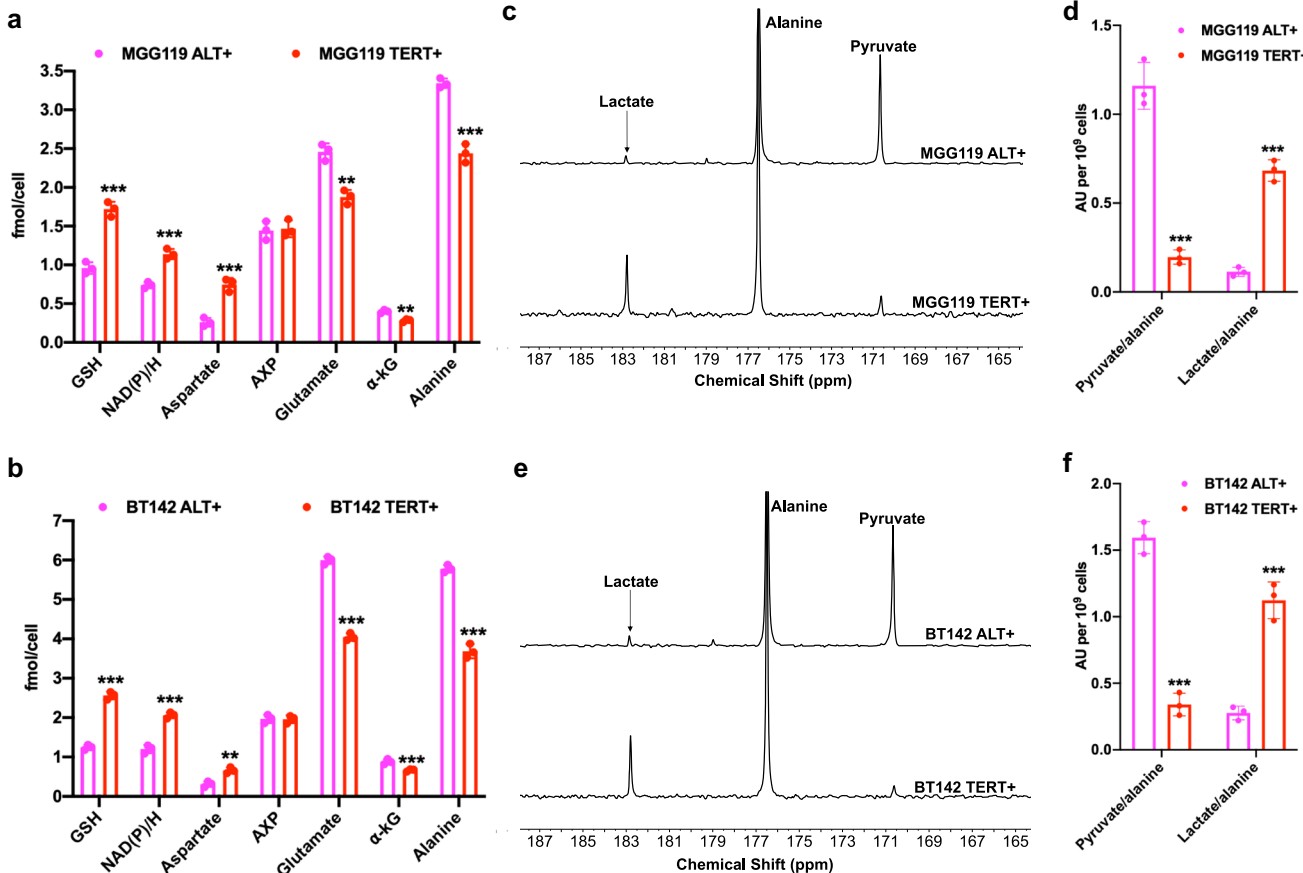

**Fig. 4 1H-MRS and hyperpolarized [1-13C]-alanine can inform on TERT expression or the ALT pathway in isogenic patient-derived low-grade glioma models. a** 1H-MRS-detectable metabolite levels in MGG119 ALT+ (magenta circles) and MGG119 TERT+ (red circles) neurospheres. **b** 1H-MRS-detectable metabolite levels in BT142 ALT+ (magenta circles) and BT142 TERT+ (red circles) neurospheres. **c** Representative summed 13C-MRS spectra showing hyperpolarized [1-13C]-alanine metabolism in MGG119 ALT+ and MGG119 TERT+ neurospheres. **d** Quantification of the ratios for hyperpolarized pyruvate/alanine and hyperpolarized lactate/alanine in MGG119 ALT+ (magenta circles) and MGG119 TERT+ (red circles) neurospheres. **e** Representative summed 13C-MRS spectra showing hyperpolarized [1-13C]-alanine metabolism in BT142 ALT+ and BT142 TERT+ neurospheres. **f** Quantification of the ratios for hyperpolarized pyruvate/alanine and hyperpolarized lactate/alanine in BT142 ALT+ (magenta circles) and BT142 TERT+ (magenta circles) neurospheres. All experiments were performed on three biological replicates ($n = 3$). Results are presented as mean ± standard deviation. Statistical significance was assessed using an unpaired Student's *t* test assuming unequal variance with $p < 0.05$ considered significant. Correction for multiple comparisons was performed using the Holm–Šídák method. *** represents statistical significance with $p < 0.005$; ** represents statistical significance with $p < 0.01$; * represents statistical significance with $p < 0.05$. Also, refer to Supplementary Fig. 4. Source data with exact $p$ values are provided as a source data file.

MGG119 TERT+ and BT142 TERT+ neurospheres relative to MGG119 ALT+ and BT142 ALT+ neurospheres, respectively. Consistent with the inability of AXP to differentiate between $NHA_{TERT}$ and $NHA_{ALT}$ models (see Supplementary Fig. 2c), there was no difference in AXP levels between MGG119 TERT+ and BT142 TERT+ neurospheres relative to their corresponding ALT+ counterparts (Fig. 4a, b). We also confirmed the increase in NADH levels in MGG119 TERT+ and BT142 TERT+ neurosphere models relative to MGG119 ALT+ and BT142 ALT+, respectively using spectrophotometric assays (Supplementary Fig. 4g, h). Importantly, consistent with elevated NADH, MGG119 TERT+ and BT142 TERT+ neurospheres showed significantly higher production of hyperpolarized [1-13C]-lactate from hyperpolarized [1-13C]-alanine relative to MGG119 ALT+ and BT142 ALT+ neurospheres, respectively (Fig. 4c–f). Collectively, the results presented in this section indicate that TERT expression and the ALT pathway are causally linked to our 1H-MRS biomarkers and hyperpolarized [1-13C]-alanine metabolism in clinically relevant, patient-derived LGOG and LGA models.

**TERT expression and the ALT pathway lead to alterations in steady-state metabolite levels in orthotopic tumor xenografts that can be detected by 1H-MRS.** Next, we examined the validity of our 1H-MRS biomarkers in orthotopic tumor xenografts. Since $NHA_{CONTROL}$ cells lack a TMM, they do not form tumors in vivo[35]. We, therefore, examined the 1H-MRS profiles of ex vivo extracts from orthotopic $NHA_{TERT}$ and $NHA_{ALT}$ tumor xenografts and compared them to tumor-free healthy controls. As shown in Fig. 5a, consistent with our cell studies, NAD(P)/H, GSH, aspartate, and AXP were significantly higher in $NHA_{TERT}$ tumors relative to normal brain and $NHA_{ALT}$ tumors. α-KG, glutamate, alanine, and AXP were significantly higher in $NHA_{ALT}$ tumors relative to normal brain and $NHA_{TERT}$ (Fig. 5a). We

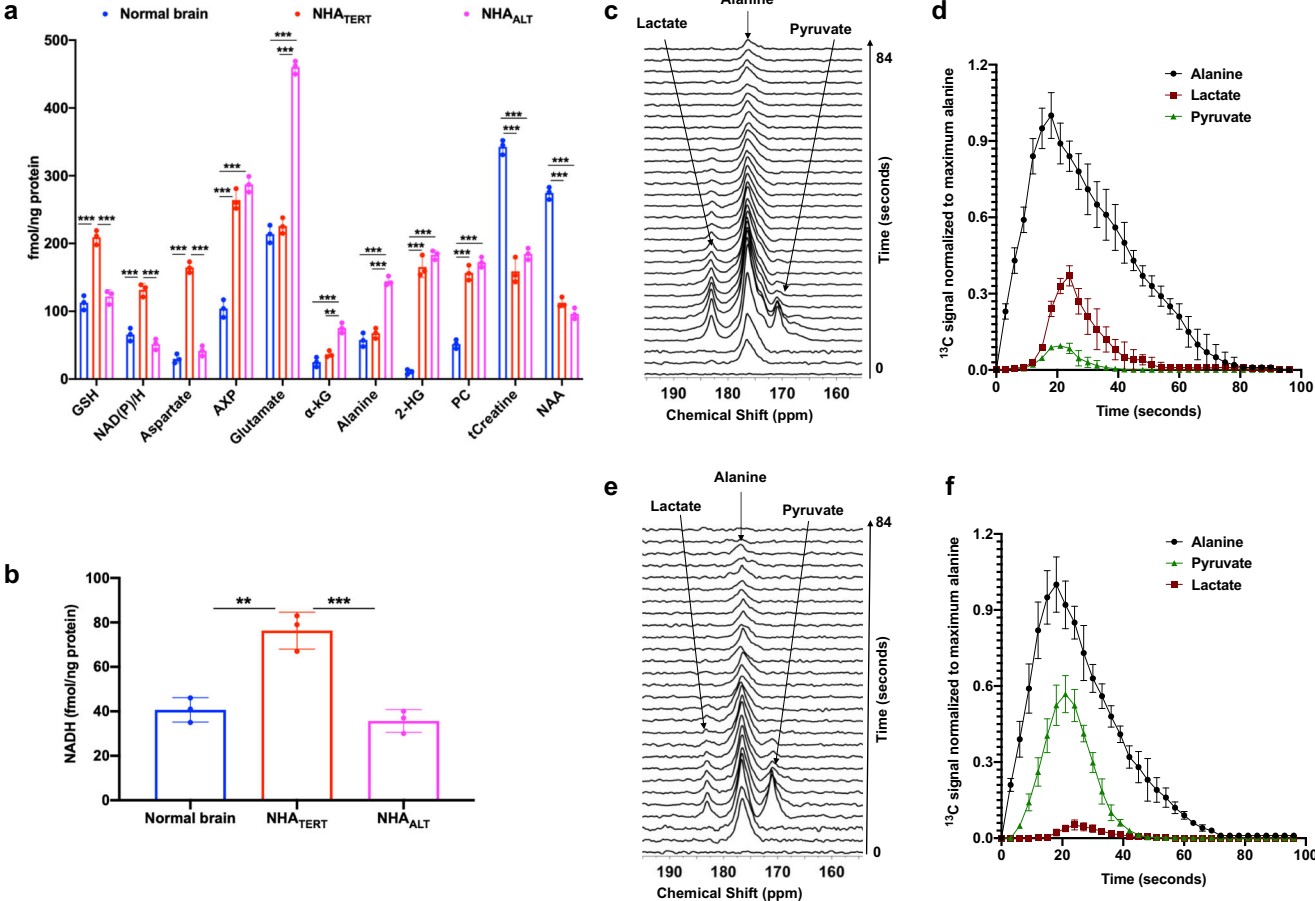

**Fig. 5 1H-MRS and hyperpolarized [1-13C]-alanine can detect TMM status in orthotopic tumor xenografts.** Quantification of steady-state metabolite levels by 1H-MRS (**a**) and NADH levels by spectrophotometry (**b**) in extracts from ex vivo tissue samples from tumor-free healthy rats (blue circles) or rats bearing either orthotopic $NHA_{TERT}$ (red circles) or $NHA_{ALT}$ (magenta circles) tumor xenografts. 2-HG: 2-hydroxyglutarate; PC: phosphocholine; tCreatine: total creatine; NAA: N-acetylaspartate. **c** Representative array of dynamic 13C-MRS spectra acquired from a 15 mm slab through the brain following intravenous injection of hyperpolarized [1-13C]-alanine into a rat bearing an orthotopic $NHA_{TERT}$ tumor. **d** Dynamic build-up curves for hyperpolarized [1-13C]-alanine (black circles), hyperpolarized [1-13C]-pyruvate (green triangles) and hyperpolarized [1-13C]-lactate (maroon squares) in rats bearing orthotopic $NHA_{TERT}$ tumor xenografts. **e** Representative 13C spectral array of hyperpolarized [1-13C]-alanine metabolism in a 15 mm slab of the brain in a rat bearing an orthotopic $NHA_{ALT}$ tumor xenograft. **f** Build-up curves for hyperpolarized [1-13C]-alanine (black circles), hyperpolarized [1-13C]-pyruvate (green triangles) and hyperpolarized [1-13C]-lactate (maroon squares) in rats bearing orthotopic $NHA_{ALT}$ tumor xenografts. All experiments were performed on three biological replicates ($n = 3$). Results are presented as mean ± standard deviation. Statistical significance was assessed using an unpaired Student's $t$ test assuming unequal variance with $p < 0.05$ considered significant. Correction for multiple comparisons was performed using the Holm–Šídák method. *** represents statistical significance with $p < 0.005$; ** represents statistical significance with $p < 0.01$. Also, see Supplementary Fig. 5. Source data with exact $p$ values are provided as a source data file.

further confirmed the significant increase in NADH in NHA$_{TERT}$ tumors by spectrophotometry (Fig. 5b). We observed metabolic changes linked to the *IDHmut* background of these models and to tumor proliferation such as significantly higher 2-HG[35] and phosphocholine[52] in both NHA$_{TERT}$ and NHA$_{ALT}$ tumors relative to normal brain (Fig. 5a). In line with previous studies[28], we also observed significantly higher total creatine and N-acetylaspartate in the normal brain relative to both tumor models (Fig. 5a). These results suggest that $^1$H-MRS has the potential to detect TERT or ALT status in orthotopic tumor xenografts.

**Hyperpolarized [1-$^{13}$C]-alanine non-invasively monitors TERT expression and the ALT pathway in genetically-engineered and patient-derived LGOG and LGA models in vivo.** We then examined the feasibility of non-invasively imaging TMM status in vivo using hyperpolarized [1-$^{13}$C]-alanine. To begin with, we acquired dynamic $^{13}$C spectra from a 15 mm axial slab of the brain and compared tumor-free healthy rats with those bearing orthotopic NHA$_{TERT}$ or NHA$_{ALT}$ tumors. Following intravenous injection of hyperpolarized [1-$^{13}$C]-alanine, we observed a build-up of [1-$^{13}$C]-pyruvate and [1-$^{13}$C]-lactate in NHA$_{TERT}$ tumor-bearing rats with lactate being the predominant product peak (Fig. 5c, d). Injection of hyperpolarized [1-$^{13}$C]-alanine into NHA$_{ALT}$ tumor-bearing rats showed considerable production of hyperpolarized [1-$^{13}$C]-pyruvate and limited further conversion to hyperpolarized [1-$^{13}$C]-lactate (Fig. 5e), making pyruvate the predominant product peak (Fig. 5e, f). In contrast, tumor-free healthy control rats did not show the metabolism of hyperpolarized [1-$^{13}$C]-alanine to either [1-$^{13}$C]-pyruvate or [1-$^{13}$C]-lactate (Supplementary Fig. 5a, b). These metabolic differences between NHA$_{TERT}$ or NHA$_{ALT}$ tumor-bearing rats and tumor-free controls are highlighted in the summed $^{13}$C spectra in Supplementary Fig. 5c. Importantly, the lactate/alanine ratio differentiated NHA$_{TERT}$ tumor from the healthy brain (Supplementary Fig. 5d), while the pyruvate/alanine ratio differentiated NHA$_{ALT}$ tumor from the healthy brain (Supplementary Fig. 5e). There was no significant difference in the SNR of hyperpolarized [1-$^{13}$C]-alanine (Supplementary Fig. 5f) between tumor-free and tumor-bearing rats. It should be noted that, in both NHA$_{TERT}$ and NHA$_{ALT}$ tumor-bearing rats, there was a temporal delay between the maxima for hyperpolarized alanine (18 s post injection) and pyruvate (21 s post injection) and a further delay between the maxima for pyruvate and lactate (24 s post injection; see Fig. 5d, f). Taken together with the lack of hyperpolarized alanine metabolism in tumor-free healthy animals (Supplementary Fig. 5a, c), these results suggest that alanine metabolism to pyruvate or lactate is likely to be originating in the tumor in the brain. If metabolism of hyperpolarized alanine were occurring in the liver or the vasculature and alanine, pyruvate, and lactate were being imported into the brain, all three species would potentially be expected to appear in the spectral array at the same time-point and to be present in healthy animals as well.

In order to further verify these findings and determine the spatial distribution of hyperpolarized [1-$^{13}$C]-alanine metabolism, we performed 2D echo-planar spectroscopic imaging (EPSI)[53]. Following intravenous injection of hyperpolarized [1-$^{13}$C]-alanine into NHA$_{TERT}$-tumor-bearing rats, production of [1-$^{13}$C]-pyruvate and [1-$^{13}$C]-lactate was observed in the tumor but not contralateral normal brain voxel and, consistent with data from cells and $^{13}$C slab acquisitions, lactate was the predominant product peak (Fig. 6a). The lactate/alanine ratio was significantly higher in the NHA$_{TERT}$ tumor voxel relative to the contralateral normal brain (Fig. 6b). There was no significant difference in the SNR of hyperpolarized [1-$^{13}$C]-alanine between tumor and contralateral normal brain

(Fig. 6c). These results are highlighted in metabolic heatmaps showing tumor-specific localization of lactate in rats bearing orthotopic NHA$_{TERT}$ tumors (Fig. 6d). In contrast, hyperpolarized [1-$^{13}$C]-alanine was distributed throughout the brain (Fig. 6d).

To establish clinical relevance, we also examined the ability of hyperpolarized [1-$^{13}$C]-alanine to assess TERT status in a patient-derived LGOG model. Since the BT54 model does not form tumors in vivo, we instead examined the SF10417 model, which was derived from a patient carrying an *IDHmut* LGOG tumor[6,54]. *TERT* expression and telomerase activity have been confirmed in the SF10417 model (Supplementary Fig. 1a, b and previously in refs. [6,54]). As shown in Fig. 6e, consistent with data from the NHA$_{TERT}$ model, production of [1-$^{13}$C]-pyruvate and [1-$^{13}$C]-lactate from hyperpolarized [1-$^{13}$C]-alanine could be observed in the tumor but not contralateral normal brain voxel, with higher lactate than pyruvate. The lactate/alanine ratio was significantly higher in the tumor voxel vs. contralateral normal brain (Fig. 6f), while there was no difference in hyperpolarized [1-$^{13}$C]-alanine SNR (Fig. 6g). Importantly, metabolic heatmaps showed that the lactate/alanine ratio delineated the tumor while alanine was distributed throughout the brain (Fig. 6h).

2D EPSI of rats bearing orthotopic NHA$_{ALT}$ tumors showed production of [1-$^{13}$C]-pyruvate and [1-$^{13}$C]-lactate in the tumor but not contralateral normal brain voxel (Fig. 7a). Consistent with data from live cells and $^{13}$C slab experiments, pyruvate was the predominant product peak and the pyruvate/alanine ratio was significantly higher in tumor vs. contralateral brain (Fig. 7b). There was no significant difference in hyperpolarized [1-$^{13}$C]-alanine SNR between tumor and contralateral normal brain (Fig. 7c). Examination of metabolic heatmaps indicated that the hyperpolarized pyruvate/alanine ratio was an effective delineator of the tumor in NHA$_{ALT}$ tumor-bearing rats (Fig. 7d).

In order to confirm our ability to non-invasively monitor ALT status in a clinically relevant LGA model, we performed 2D EPSI on rats bearing orthotopic BT142 tumor xenografts. As shown in Fig. 7e, consistent with data from NHA$_{ALT}$ tumors, hyperpolarized [1-$^{13}$C]-alanine metabolism to [1-$^{13}$C]-pyruvate and [1-$^{13}$C]-lactate was observed in the tumor voxel but not the contralateral normal brain, with pyruvate being the predominant product peak. The pyruvate/alanine ratio was significantly higher in the BT142 tumor relative to the contralateral normal brain (Fig. 7f) with no significant difference in hyperpolarized [1-$^{13}$C]-alanine SNR (Fig. 7g). These results were highlighted in metabolic heatmaps showing demarcation of the tumor by the pyruvate/alanine ratio (Fig. 7h). Taken together, these results validate the utility of hyperpolarized [1-$^{13}$C]-alanine in non-invasively visualizing TMM status in genetically-engineered and patient-derived LGOG and LGA models in vivo.

**Glutaminase 1 (GLS1), nicotinamide phosphoribosyl transferase (NAMPT), and the alanine transporters ASCT2 and LAT2 are molecular determinants of TMM-linked metabolic alterations.** Next, we set out to mechanistically validate our metabolic biomarkers. GLS1 is a key component in α-KG homeostasis[31]. GLS1 is the rate-limiting enzyme in the conversion of glutamine to glutamate which is in equilibrium with α-KG[31]. GLS activity was significantly higher in NHA$_{ALT}$ cells relative to NHA$_{TERT}$ and NHA$_{CONTROL}$ (Supplementary Fig. 6a). Examination of ex vivo tumor tissues showed that GLS activity was also significantly higher in NHA$_{ALT}$ tumors relative to NHA$_{TERT}$ and tumor-free normal brain (Supplementary Fig. 6b). Importantly, GLS activity was significantly reduced in BT142 ALT− neurospheres relative to BT142 ALT+ (Fig. 8a). In order to directly confirm the role of GLS1 in linking ALT to α-KG and hyperpolarized [1-$^{13}$C]-alanine metabolism, we silenced *GLS1* in BT142 neurospheres by RNA

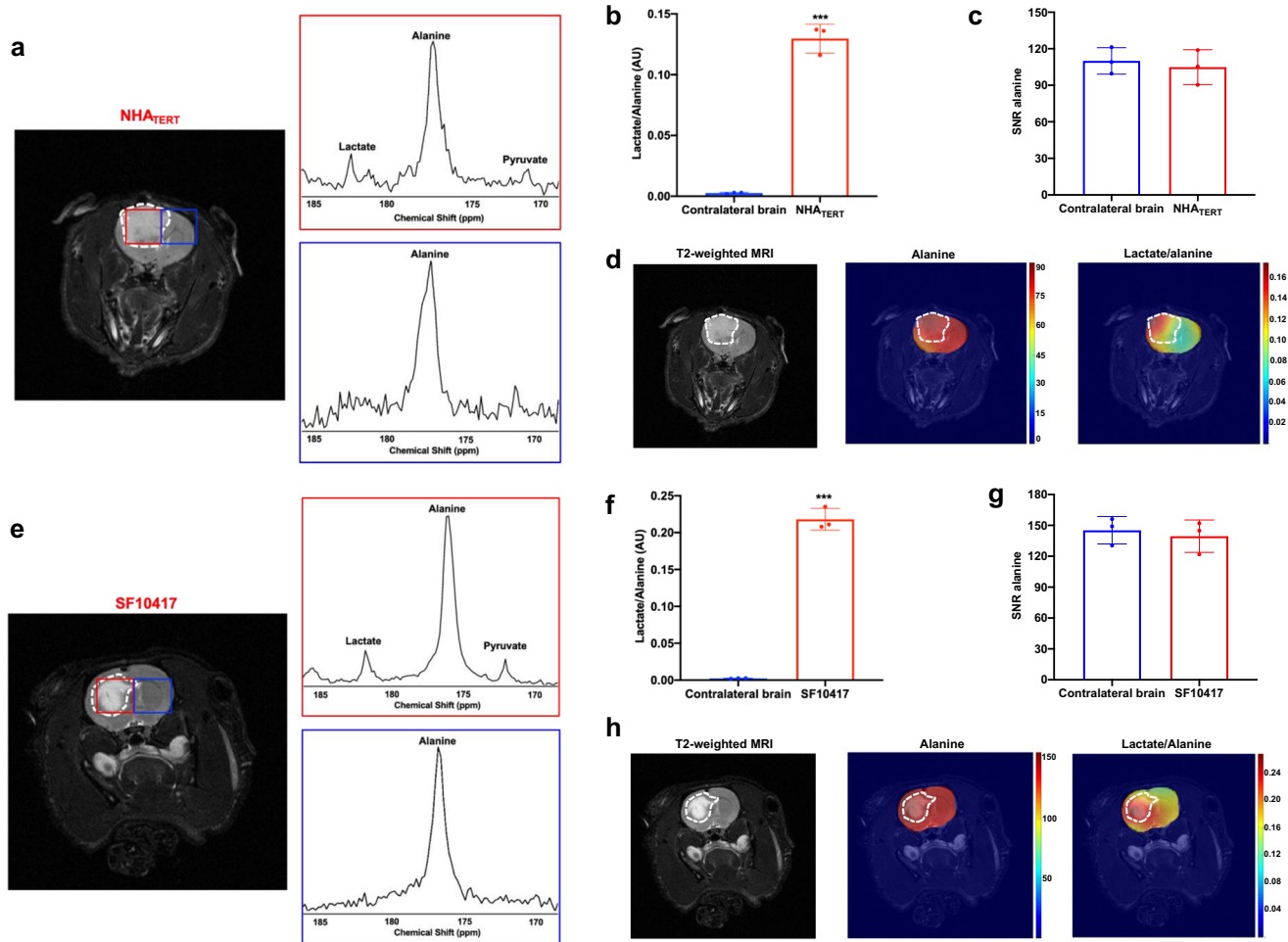

**Fig. 6 Hyperpolarized [1-¹³C]-alanine metabolism to lactate is a non-invasive imaging biomarker of TERT status in both genetically-engineered and patient-derived LGOG models in vivo. a** 2D EPSI of hyperpolarized [1-¹³C]-alanine metabolism to predominantly lactate in a rat bearing an orthotopic NHA_TERT tumor. Tumor voxel is indicated in red and the contralateral normal brain voxel in blue. Quantification of the hyperpolarized lactate/alanine ratio (**b**) and the SNR of hyperpolarized [1-¹³C]-alanine (**c**) in the tumor (red) and contralateral normal brain (blue) voxels in orthotopic NHA_TERT tumors. **d** Representative heatmaps of hyperpolarized [1-¹³C]-alanine metabolism in a rat bearing an orthotopic NHA_TERT tumor xenograft. Left panel: T2-weighted MRI with the tumor region contoured in white. Middle panel: Heatmap of hyperpolarized [1-¹³C]-alanine distribution; Right panel: a heatmap of the hyperpolarized lactate/alanine ratio. **e** 2D EPSI of hyperpolarized [1-¹³C]-alanine metabolism to predominantly lactate in a rat bearing an orthotopic SF10417 tumor xenograft. Tumor voxel is indicated in red and the contralateral normal brain voxel in blue. Hyperpolarized lactate/alanine ratio (**f**) and the SNR of hyperpolarized [1-¹³C]-alanine (**g**) in the tumor (red) voxel vs. contralateral normal brain (blue) voxel in orthotopic SF10417 tumors. **h** Representative heatmaps of hyperpolarized [1-¹³C]-alanine metabolism in a rat bearing an orthotopic SF10417 tumor xenograft. Left panel: T2-weighted MRI with the tumor region contoured in white. Middle panel: Heatmap of hyperpolarized [1-¹³C]-alanine distribution; Right panel: a heatmap of the ratio of hyperpolarized lactate/alanine. All experiments were performed on three biological replicates ($n = 3$). Data are presented as mean ± standard deviation. Statistical significance was assessed using an unpaired Student's $t$ test assuming unequal variance with $p < 0.05$ considered significant. *** represents statistical significance with $p < 0.005$. Source data with exact $p$ values are provided as a source data file.

interference using two nonoverlapping siRNA oligonucleotides. We confirmed a significant loss of GLS1 expression and activity in BT142 GLS− neurospheres (Supplementary Fig. 6c, d). ¹H-MRS-detectable glutamate and α-KG were significantly reduced in BT142 GLS− neurospheres relative to BT142 GLS+ (Fig. 8b). Examination of hyperpolarized [1-¹³C]-alanine metabolism showed a significant reduction in the production of both pyruvate and lactate (Fig. 8c), thereby leading to significantly reduced pyruvate/alanine and lactate/alanine ratios in BT142 GLS− neurospheres relative to BT142 GLS+ (Supplementary Fig. 6e). These results identify GLS1 as a mechanistic link between the ALT pathway, α-KG, and hyperpolarized [1-¹³C]-alanine metabolism to pyruvate.

NAMPT is the rate-limiting enzyme that converts nicotinamide to nicotinamide mononucleotide and plays an essential role in

NAD⁺ biosynthesis in cancer[55], including low-grade gliomas[56]. Consistent with elevated NAD⁺ and NADH in NHA_TERT cells (see Fig. 1d, e), NAMPT activity was significantly higher in NHA_TERT cells relative to NHA_ALT and NHA_CONTROL (Supplementary Fig. 6f). NAMPT activity was also significantly higher in NHA_TERT tumor tissue relative to NHA_ALT and tumor-free healthy brain (Supplementary Fig. 6g). Importantly, *TERT* silencing significantly reduced NAMPT activity in BT54 neurospheres (Fig. 8d). To interrogate the role of NAMPT in TERT cells, we silenced *NAMPT* in BT54 neurospheres by RNA interference using two nonoverlapping siRNA sequences and confirmed the significant loss of NAMPT expression and activity (Supplementary Fig. 6h, i). *NAMPT* silencing significantly reduced NADH (Fig. 8e) and NAD⁺ (Supplementary Fig. 6j). Furthermore, [1-¹³C]-lactate production from hyperpolarized [1-¹³C]-alanine was significantly

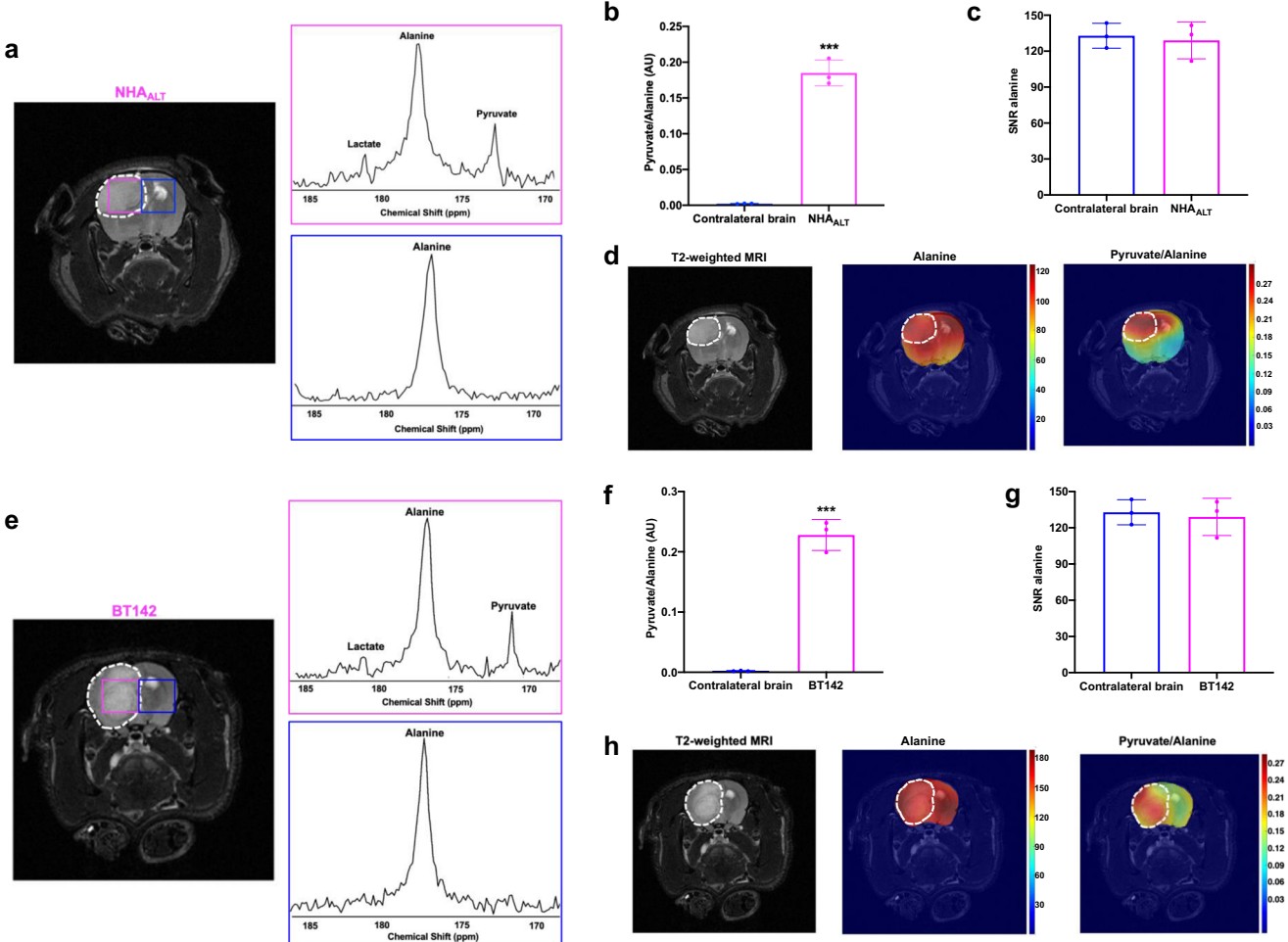

**Fig. 7 Hyperpolarized [1-$^{13}$C]-alanine metabolism to pyruvate is a non-invasive imaging biomarker of ALT status in both genetically-engineered and patient-derived LGA models in vivo. a** 2D EPSI of hyperpolarized [1-$^{13}$C]-alanine metabolism to predominantly pyruvate in a rat bearing an orthotopic NHA$_{ALT}$ tumor. Tumor voxel is indicated in magenta and the contralateral normal brain voxel in blue. Hyperpolarized pyruvate/alanine ratio (**b**) and hyperpolarized [1-$^{13}$C]-alanine SNR (**c**) in the tumor (magenta) and contralateral normal brain (blue) voxels in orthotopic NHA$_{ALT}$ tumors. **d** Representative heatmaps of hyperpolarized [1-$^{13}$C]-alanine metabolism in a rat bearing an orthotopic NHA$_{ALT}$ tumor xenograft. Left panel: T2-weighted MRI with the tumor region contoured in white. Middle panel: a heatmap of hyperpolarized [1-$^{13}$C]-alanine distribution; Right panel: a heatmap of the hyperpolarized pyruvate/alanine ratio. **e** 2D EPSI of hyperpolarized [1-$^{13}$C]-alanine metabolism to predominantly pyruvate in a rat bearing an orthotopic BT142 tumor xenograft. Tumor voxel is indicated in magenta and the contralateral normal brain voxel in blue. Quantification of the hyperpolarized pyruvate/alanine ratio (**f**) and the SNR of hyperpolarized [1-$^{13}$C]-alanine (**g**) in the tumor (magenta) voxel vs. contralateral normal brain (blue) voxel in orthotopic BT142 tumors. **h** Representative heatmaps of hyperpolarized [1-$^{13}$C]-alanine metabolism in a rat bearing an orthotopic BT142 tumor xenograft. Left panel: T2-weighted MRI with the tumor region contoured in white. Middle panel: a heatmap of hyperpolarized [1-$^{13}$C]-alanine distribution; Right panel: a heatmap of the ratio of hyperpolarized pyruvate/alanine. All experiments were performed on three biological replicates ($n = 3$). Results are presented as mean ± standard deviation. Statistical significance was assessed using an unpaired Student's $t$ test assuming unequal variance with $p < 0.05$ considered significant. *** represents statistical significance with $p < 0.005$. Source data with exact $p$ values are provided as a source data file.

reduced in BT54 NAMPT− neurospheres relative to BT54 NAMPT+, causing an accumulation of [1-$^{13}$C]-pyruvate (Fig. 8f and Supplementary Fig. 6k). These results point to a mechanistic role for NAMPT in linking TERT to NADH and to hyperpolarized [1-$^{13}$C]-alanine metabolism to lactate.

We also examined other factors involved in hyperpolarized [1-$^{13}$C]-alanine metabolism (see schematic in Fig. 2a). Previous studies suggest that alanine is transported across the blood–brain–barrier (BBB) and into the cell via the amino acid transporters alanine, serine, cysteine transporter 2 (ASCT2), and the L-type amino acid transporter (LAT2)[57–59]. *ASCT2* and *LAT2* expression were higher in both NHA$_{TERT}$ and NHA$_{ALT}$ cells relative to NHA$_{CONTROL}$ (Supplementary Fig. 7a). In line with higher transporter expression, intracellular import of [1-$^{13}$C]-

alanine from the medium was significantly higher in NHA$_{TERT}$ and NHA$_{ALT}$ cells relative to NHA$_{CONTROL}$ (Supplementary Fig. 7b). Importantly, silencing *TERT* expression significantly reduced *ASCT2* and *LAT2* expression and [1-$^{13}$C]-alanine import in the BT54 LGOG model (Fig. 8g, h). Similarly, expression of *ASCT2* and *LAT2* as well as [1-$^{13}$C]-alanine import were significantly reduced in BT142 ALT- neurospheres relative to BT142 ALT+ (Fig. 8i, j). These results suggest that both TERT expression and the ALT pathway are mechanistically associated with elevated alanine import in LGOG and LGAs, respectively.

Examination of alanine aminotransferase, which converts alanine to pyruvate, showed no significant differences in activity between NHA$_{TERT}$, NHA$_{ALT}$, and NHA$_{CONTROL}$ cells (Supplementary Fig. 7c). No significant differences in alanine aminotransferase

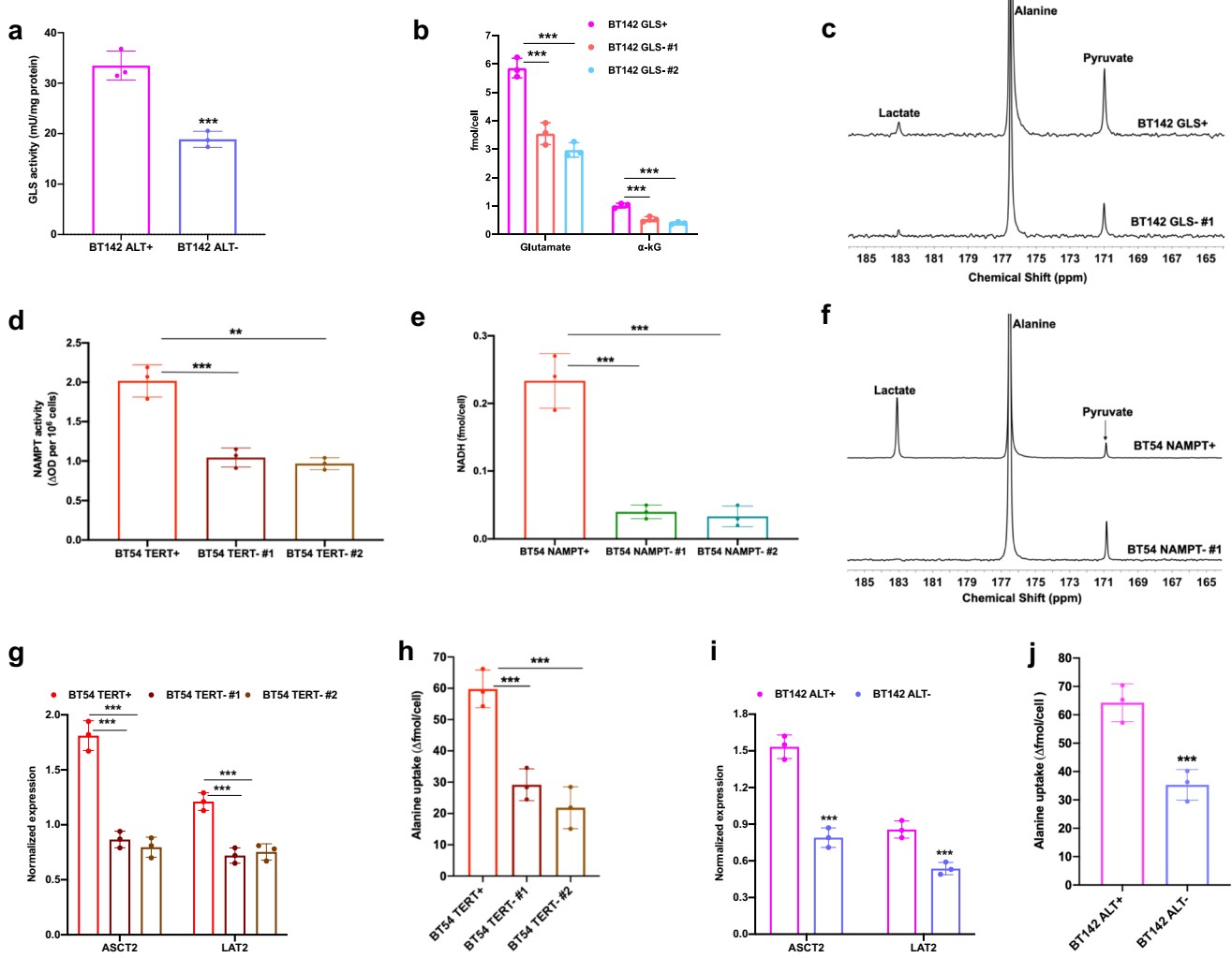

**Fig. 8 GLS1, NAMPT, ASCT2, and LAT2 mechanistically link TERT expression and the ALT pathway to metabolic reprogramming in patient-derived LGOG and LGA models. a** GLS activity in BT142 ALT+ (magenta circles) and BT142 ALT− (lavender circles) neurospheres. **b** Effect of *GLS1* silencing on ¹H-MRS-detectable levels of glutamate and α-KG in the BT142 model (BT142 GLS+ neurospheres: magenta circles; BT142 GLS− #1 neurospheres: orange circles; BT142 GLS− #2 neurospheres: cyan circles). *GLS1* silencing was achieved using two independent, nonoverlapping siRNA sequences/pools. **c** Representative summed ¹³C spectra showing hyperpolarized [1-¹³C]-alanine metabolism in BT142 GLS+ and BT142 GLS− neurospheres. **d** NAMPT activity in BT54 TERT+ (red circles), BT54 TERT− #1 (dark brown circles), and BT54 TERT− #2 (light brown circles) neurospheres. *TERT* silencing was carried out using two independent, nonoverlapping siRNA sequences/pools. **e** NADH levels measured by spectrophotometry in BT54 NAMPT+ (red circles), BT54 NAMPT− #1 (green circles), and BT54 NAMPT− #2 (teal circles) neurospheres. *NAMPT* silencing was achieved using two independent, nonoverlapping siRNA sequences/pools. **f** Representative summed ¹³C spectra showing hyperpolarized [1-¹³C]-alanine metabolism in BT54 NAMPT+ and BT54 NAMPT− neurospheres. **g** Expression of *ASCT2* and *LAT2* in BT54 TERT+ (red circles), BT54 TERT− #1 (dark brown circles), and BT54 TERT− #2 (light brown circles) neurospheres. *TERT* was silenced using two independent, nonoverlapping siRNA sequences/pools. **h** Effect of *TERT* silencing on intracellular import of [1-¹³C]-alanine as measured by thermally-polarized ¹³C-MRS in BT54 TERT+ (red circles), BT54 TERT− #1 (dark brown circles), and BT54 TERT− #2 (light brown circles) neurospheres. (**i**) *ASCT2* and *LAT2* expression in BT142 ALT+ (magenta circles) and BT142 ALT− (lavender circles) neurospheres. **j** [1-¹³C]-alanine import as measured by thermally polarized ¹³C-MRS in BT142 ALT+ (magenta circles) and ALT− (lavender circles) neurospheres. All experiments were performed on three biological replicates ($n = 3$). Results are presented as mean ± standard deviation. Statistical significance was assessed using an unpaired Student's *t* test assuming unequal variance with $p < 0.05$ considered significant. *** represents statistical significance with $p < 0.005$. ** represents statistical significance with $p < 0.01$. Also, see Supplementary Fig. 6. Source data with exact p values are provided as a source data file.

activity were observed between normal brain and NHA_TERT or NHA_ALT tumors (Supplementary Fig. 7d). Lactate to pyruvate conversion depends on lactate dehydrogenase activity and steady-state levels of NADH[40] and lactate[41]. There were no significant differences in lactate dehydrogenase activity or steady-state lactate between NHA_TERT, NHA_ALT, and NHA_CONTROL cells or between normal brain and NHA_TERT or NHA_ALT tumors (Supplementary Fig. 7e–h). Taken together, the results presented in this section identify mechanistic roles for GLS1, NAMPT, ASCT2, and LAT2 in

hyperpolarized [1-¹³C]-alanine metabolism to pyruvate and to lactate in ALT and TERT models, respectively.

**TERT expression and the ALT pathway are linked to MRS-detectable metabolic reprogramming in LGOG and LGA patient biopsies.** Finally, in order to establish the validity of our metabolic biomarkers to human patients, we examined biopsies from LGOG and LGA patients. Since obtaining normal brain

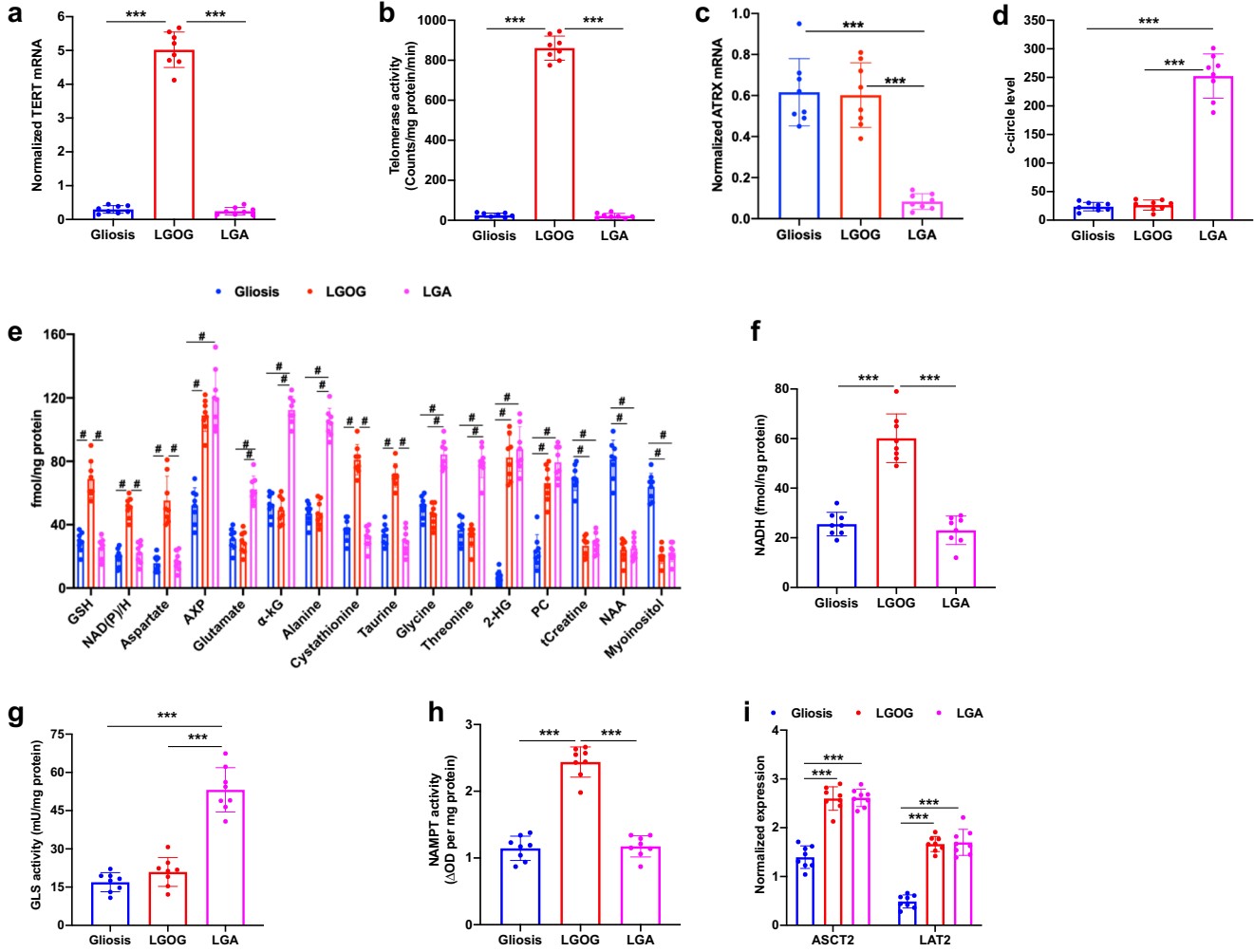

**Fig. 9 TERT expression and the ALT pathway are associated with MRS-detectable metabolic reprogramming in LGOG and LGA patient biopsies.** *TERT* mRNA expression (**a**), telomerase activity (**b**), *ATRX* mRNA expression (**c**), and c-circle level (**d**) in LGOG (red circles), LGA (magenta circles), and gliosis (blue circles) biopsies. Steady-state metabolite levels measured by $^1$H-MRS (**e**) and NADH levels measured by spectrophotometry (**f**) in ex vivo extracts from LGOG (red circles), LGA (magenta circles), and gliosis (blue circles) biopsies. 2-HG: 2-hydroxyglutarate; PC: phosphocholine; tCreatine: total creatine; NAA: N-acetylaspartate. GLS activity (**g**), NAMPT activity (**h**), and expression of *ASCT2* and *LAT2* (**i**) in LGOG (red circles), LGA (magenta circles), and gliosis (blue circles) biopsies. All experiments were performed on eight biological replicates each ($n = 8$). Results are presented as mean ± standard deviation. Statistical significance was assessed using an unpaired Student's $t$ test assuming unequal variance with $p < 0.05$ considered significant. Correction for multiple comparisons was performed using the Holm–Šídák method. *** represents statistical significance with $p < 0.005$. For the sake of clarity, statistical significance with $p < 0.005$ is represented with # for panel (**e**). Source data with exact $p$ values are provided as a source data file.

biopsies is challenging, we compared tumor tissues to non-neoplastic gliosis biopsies as described previously[60]. This also serves the purpose of testing whether monitoring TMM status can differentiate between tumor and gliosis, which is currently a challenging clinical problem[24,25]. First, we confirmed *TERT* expression as well as telomerase activity in LGOG biopsies (Fig. 9a, b). We confirmed the presence of the ALT pathway in LGA biopsies by measuring *ATRX* loss and c-circle formation (Fig. 9c, d). Consistent with data from our LGOG models, NAD (P)/H, GSH, aspartate, and AXP were significantly elevated in LGOG biopsies relative to LGA and gliosis (Fig. 9e). We confirmed the increase in NADH by spectrophotometric assays (Fig. 9f). In line with previous studies, we also observed significantly elevated cystathionine[61] and taurine[62] in LGOG biopsies relative to LGA and gliosis. α-KG, glutamate, alanine, and AXP were significantly higher in LGA biopsies relative to LGOG and gliosis (Fig. 9e), consistent with our cell and ex vivo data from LGA models. In line with previous reports, glycine, and threonine were also elevated in LGA biopsies relative to LGOG

and gliosis[61]. As expected, 2-HG and phosphocholine were elevated in both LGOG and LGA biopsies relative to gliosis[62]. Gliosis samples had significantly higher myoinositol, total creatine, and N-acetylaspartate relative to LGOG and LGA. Importantly, consistent with data from our cell and ex vivo models, GLS activity was significantly higher in LGA biopsies relative to LGOG and gliosis (Fig. 9g), NAMPT activity was significantly higher in LGOG biopsies relative to LGA and gliosis (Fig. 9h) and expression of the alanine transporters *ASCT2* and *LAT2* were higher in both LGOG and LGA biopsies relative to gliosis (Fig. 9i). Collectively, these results mechanistically validate our MRS-detectable metabolic biomarkers and putative molecular mechanisms in patient samples and highlight their potential utility in distinguishing tumor from gliosis.

## Discussion

Telomere maintenance is a fundamental hallmark of cancer[1]. The majority (~85%) of human tumors, including the LGOGs, use TERT expression for telomere maintenance[2–5], while ~15% of

tumors, including the LGAs, use the ALT pathway[8–10]. Due to their inherent link to tumor proliferation, imaging TERT expression or the ALT pathway has the potential to differentiate tumor from the normal brain and anatomically similar lesions of gliosis, edema, and necrosis, and to discriminate treatment response from pseudoprogression or pseudoresponse. TERT expression and the ALT pathway are also therapeutic targets[2,9,63], and non-invasive methods of imaging TERT or ALT status can aid in the development and future clinical translation of TERT or ALT inhibitors. However, non-invasive methods of imaging TERT or ALT status are currently lacking. In this study, we have, for the first time, leveraged an understanding of the metabolic consequences of TERT expression and the ALT pathway to identify combinations of [1H]-MRS biomarkers with differential hyperpolarized [1-13C]-alanine metabolism that can non-invasively inform on TMM status in vivo.

Previous studies have identified noncanonical roles for TERT in NF-κB and Wnt/β-catenin signaling, apoptotic resistance, and metabolic reprogramming[16]. Our findings strengthen the link between TERT and metabolic reprogramming. Specifically, our findings of elevated GSH in patient-derived LGOG models and LGOG patient biopsies reinforce previous studies in genetically-engineered model systems linking TERT to elevated GSH and NADPH[17–19]. Our study also identifies novel metabolic alterations, such as increased aspartate, NAD+, and NADH that have not previously been linked to TERT expression and could potentially serve as metabolic imaging or therapeutic targets. Importantly, our study is the first to link the ALT pathway with metabolic reprogramming in cancer.

Diffuse gliomas have traditionally been classified by histological grade. However, histopathologically similar gliomas carry different oncogenic mutations driving tumor initiation and maintenance[64]. As a result, the World Health Organization 2016 classification scheme includes molecular features to guide clinical decision-making[64]. Among the molecular features that define low-grade gliomas, mutually exclusive mutations in the TERT promoter and ATRX, which define LGOGs and LGAs, respectively[4,10,64], are notable because they both lead to TMMs. However, non-invasive methods of detecting TMM status are lacking. Currently, detection of TMM status is based on analysis of tumor biopsies[2,8,37,38]. However, biopsy sampling is invasive, difficult, and restricts longitudinal analysis of TMM status in response to treatment or during tumor recurrence[22,23]. While PET or optical imaging methods for detection of TERT expression have been reported[65,66], they require the use of antibodies[65] or anti-sense oligonucleotides[66] that restricts application to gliomas due to limited BBB penetration. Moreover, PET imaging is radioactive, which limits longitudinal assessment of TMM status due to concerns of radiation exposure, while the application of optical imaging to gliomas is limited by depth penetration[67]. In contrast, the [1H]- and hyperpolarized [13C]-MRS biomarkers reported here provide non-invasive, clinically translatable methods of imaging TMM status in a longitudinal manner during disease progression and treatment response. Importantly, our biomarkers provide a way of integrating our understanding of glioma genetics with non-invasive imaging modalities that can impact clinical patient management.

Multiple lines of evidence suggest that our metabolic biomarkers are ultimately linked to TMM status. First, the different TMMs, i.e., TERT expression and the ALT pathway have different metabolic signatures, suggesting that the metabolic alterations are not simply the result of telomere maintenance-mediated tumor proliferation. This is further emphasized by the observation that traditional markers of tumor proliferation such as phosphocholine[52], are not linked to TMM status. Second, silencing TERT expression or the ALT pathway normalizes their

associated [1H]-MRS-detectable biomarkers as well as hyperpolarized [1-13C]-alanine metabolism, pointing to a causal link between TMMs and these biomarkers. Similarly, exogenous TERT expression in isogenic ALT+ models modulates [1H]-MRS-detectable levels of TERT− and ALT-linked metabolites as well as hyperpolarized [1-13C]-alanine metabolism. Furthermore, ATRX-deficient NHAs that do not express IDHmut and have previously been shown to lack the ALT pathway[13] do not display the metabolic alterations associated with the ALT phenotype, further pointing to the specificity of our MRS biomarkers. Third, we have identified NAMPT, GLS1, and the alanine transporters ASCT2 and LAT2 as mechanistic links between TERT expression and the ALT pathway and their associated [1H]- and hyperpolarized [13C]-MRS-detectable metabolic biomarkers. Finally, we find that hyperpolarized [1-13C]-alanine is able to non-invasively monitor TMM status in a vast array of genetically-engineered and patient-derived LGOG and LGA models. This concordance highlights the robustness of hyperpolarized [1-13C]-alanine metabolism for TMM imaging and suggests that, although we cannot fully rule out the possibility that other molecular factors affect our metabolic biomarkers, nevertheless, TERT expression and the ALT pathway are dominant factors.

Our study points to the potential utility of [1H]-MRS for imaging TMM status. Previous studies have reported the detection of GSH in glioma patients in vivo[68,69]. We also detected elevated levels of the GSH precursor cystathionine in LGOG patient biopsies, consistent with a previous study reporting cystathionine as an in vivo biomarker of LGOGs[61]. Elevated glutamate in LGAs can also be detected by [1H]-MRS in vivo[68]. Importantly, in lieu of a single metabolic biomarker, we propose the use of distinct sets of [1H]-MRS-detectable biomarkers which, in combination with differential hyperpolarized [1-13C]-alanine metabolism, have the potential to provide a non-invasive "fingerprint" of TMM status in low-grade gliomas.

Lactate dehydrogenase A is underexpressed in low-grade gliomas as a result of promoter methylation[70]. Nevertheless, lactate dehydrogenase activity and pyruvate flux to lactate are detectable in our patient-derived LGOG and LGA models, possibly due to residual lactate dehydrogenase A expression that is below immunoblotting detection limits, variations in lactate dehydrogenase A promoter methylation[71] or lactate dehydrogenase B expression[72]. Importantly, when comparing LGOG and LGA models, the difference in lactate production from hyperpolarized [1-13C]-alanine results from elevated NADH in LGOGs since reducing NADH production via NAMPT silencing abrogates lactate production and leads to accumulation of pyruvate.

In theory, hyperpolarized [1-13C]-pyruvate metabolism to lactate should also be useful to monitor the TERT-linked increase in NADH in LGOGs[40]. However, a previous study demonstrated that hyperpolarized [1-13C]-lactate production from pyruvate was not significantly higher in tumors relative to the contralateral normal brain in the BT142 LGA model[73]. Our results pointing to high pyruvate production from alanine but limited further lactate production in the BT142 model as a result of its ALT status are consistent with this prior study and serve to highlight the utility of hyperpolarized [1-13C]-alanine. Ruiz-Rodado et al.[71] also showed that lactate production from hyperpolarized [1-13C]-pyruvate was limited in BT142 tumors, but significantly higher in the tumor vs. contralateral normal brain in TS603 and NCH1681 models. They linked these differences to tumor aggressiveness and varying levels of lactate dehydrogenase A silencing and suggest that hyperpolarized lactate production from pyruvate is a biomarker of aggressiveness in low-grade gliomas[71]. Our results suggest that the TMM status of low-grade glioma models might be another contributing factor. It is also possible that differences

in the utility of hyperpolarized alanine and pyruvate stem from differences in expression of their respective transporters. Transporter expression critically influences the efficacy of hyperpolarized $^{13}$C imaging agents as evidenced by a recent study indicating that expression of the pyruvate transporters MCT1 and MCT4 is rate-limiting for hyperpolarized [1-$^{13}$C]-pyruvate flux to lactate in vivo[74]. In this context, it is important to note that the expression of MCT1 and MCT4 is silenced in low-grade gliomas, potentially as a result of IDHmut-mediated promoter hypermethylation[70,75]. In contrast, our results indicate that TERT expression and the ALT pathway are mechanistically linked to higher expression of the alanine transporters ASCT2 and LAT2 and higher alanine import in patient-derived LGOG and LGA models and patient biopsies, consistent with prior studies point to high expression of ASCT2 and LAT2 in low-grade gliomas[57–59]. These results serve to highlight the potential utility of hyperpolarized [1-$^{13}$C]-alanine for metabolic imaging of LGOGs and LGAs.

To the best of our knowledge, this is the first demonstration of the use of hyperpolarized [1-$^{13}$C]-alanine for tumor imaging in vivo. Hyperpolarized [1-$^{13}$C]-alanine has been used to probe NADH in rat body and liver[42,43], but not in cancer. In terms of translational potential, hyperpolarized [1-$^{13}$C]-alanine is neutral, BBB-permeable[57,76], has a sufficiently long T1 (~44 s) at clinically relevant field strength, and is non-toxic at the doses used in our study. Importantly, as mentioned above, the alanine transporters ASCT2 and LAT2 are highly expressed in LGOGs and LGAs[57–59]. Another advantage of hyperpolarized [1-$^{13}$C]-alanine is the ability to measure two metabolic products i.e. pyruvate and lactate[42,43]. Our studies with hyperpolarized [1-$^{13}$C]-alanine in TERT- neurospheres indicate that silencing TERT expression abrogates hyperpolarized lactate production and leads to an accumulation of hyperpolarized pyruvate. Conversely, TERT expression in ALT+ models leads to elevated lactate production from hyperpolarized alanine and a corresponding drop in hyperpolarized pyruvate. Therefore, the ability to visualize alanine metabolism to two products, i.e., pyruvate or lactate could be considered an advantage.

Due to their essential role, TMMs are therapeutic targets[2,6,9,14]. Our results indicate that silencing TMMs normalizes $^1$H-MRS-detectable biomarkers and hyperpolarized [1-$^{13}$C]-alanine metabolism. Importantly, this normalization occurs as early as 72 h, without accompanying impacts on cell proliferation. These results are consistent with previous studies indicating the existence of a lag period following TERT or ALT inhibition, prior to the onset of cell death due to telomere loss[11,77]. Thus, our metabolic biomarkers have the potential to provide an early readout of response to TMM inhibitors and, thereby, aid in their clinical translation. Our studies also suggest that $^1$H-MRS and hyperpolarized [1-$^{13}$C]-alanine have the potential to monitor the development of resistance to TMM inhibitors. Previous studies indicate that TERT inhibition could lead to the development of resistance via the ALT pathway in other cancers[78,79], although studies in gliomas are lacking. Likewise, it is possible that inhibition of the ALT pathway could lead to the development of resistance via activation of TERT expression[63]. In this context, we performed studies with isogenic patient-derived models (MGG119 and BT142) that were engineered to be either ALT+ or TERT+. Although we confirmed the TERT+ or ALT+ phenotypes of these models using a variety of methods (corroboration of TERT mRNA levels and telomerase activity for TERT status and verification of c-circle formation, T-SCE, and ATRX expression for ALT status), we cannot fully rule out the possibility that both TERT and ALT phenotypes co-exist within the same cells as reported in previous studies[48–50]. Nevertheless, our results

indicate that ALT+ neurospheres can be differentiated from TERT+ neurospheres via assessment of $^1$H-MRS-detectable metabolite levels and hyperpolarized [1-$^{13}$C]-alanine metabolism.

A current challenge in glioma imaging is the inability of anatomical MRI to distinguish "true" tumor from morphologically similar areas of gliosis, edema, and necrosis[25]. Another major challenge is pseudoprogression, which is defined as the occurrence of treatment-related effects that mimic tumor recurrence by anatomical MRI[24,26]. In contrast, treatment with anti-angiogenic agents such as bevacuzimab results in pseudoresponse due to an apparent decrease in contrast enhancement resulting from reduced tumor permeability while tumor progression is actually in effect[24,26]. The consequences of an incorrect assessment are unnecessary/delayed treatment, morbidity, and shortened survival. In this context, our studies indicate that $^1$H-MRS biomarkers of TERT expression, NAMPT activity, and ASCT2 and LAT2 expression are higher in LGOG biopsies relative to gliosis while $^1$H-MRS biomarkers of the ALT pathway, GLS activity, ASCT2, and LAT2 expression are higher in LGA biopsies relative to gliosis. These results highlight the potential of assessing TMM status in differentiating between tumor and morphologically similar lesions such as gliosis. Further studies are needed to fully assess the ability of imaging TMM status to report on pseudoprogression and pseudoresponse.

Our results identify $^1$H- and hyperpolarized $^{13}$C-MRS-detectable metabolic biomarkers of TERT expression and the ALT pathway in LGOGs and LGAs. Our studies with NHAs that were engineered to express TERT or to be ATRX-deficient in the absence of IDHmut suggest these $^1$H-MRS-detectable biomarkers are observed in the context of IDHmut-expressing low-grade gliomas. Nevertheless, our approach has broad applicability since it shows the feasibility of imaging TMMs, which are universally required for tumor proliferation. In this context, it is important to note that TERT expression is required for the proliferation of most cancer types including hepatocellular carcinomas, metastatic melanomas, and high-grade primary glioblastomas[2–5]. In contrast, pediatric high-grade gliomas and pancreatic neuroendocrine tumors use the ALT pathway as their TMM[4,8,9]. Further studies are needed to examine the utility of our metabolic imaging approach and biomarkers in these tumor types.

In summary, we have identified and mechanistically validated unique combinations of $^1$H- and hyperpolarized $^{13}$C-MRS-detectable imaging biomarkers of TMM status in glioma cell models, orthotopic tumor xenografts, and patient biopsies. Our metabolic imaging biomarkers enable non-invasive, longitudinal detection of a molecular hallmark of tumor proliferation and have the potential to impact diagnosis, stratification, and treatment response monitoring for patients with low-grade gliomas, and potentially, other cancers.

## Methods

**Cell culture**. A comprehensive summary of cell lines used in this study and the verification of their TMM status is presented in Supplementary Table 1. Genetically-engineered models: Generation and characterization of NHA$_{CONTROL}$, NHA$_{TERT}$, NHA$_{ALT}$, NHA$_{TERT+IDHmut-}$ and NHA$_{ATRX- IDHmut-}$ cells have previously been described[13,32,35,36]. Briefly, NHAs immortalized via loss of p53 and pRb were engineered to express IDHmut to generate the NHA$_{CONTROL}$ cell line[35]. To generate NHA$_{TERT}$ cells, NHA$_{CONTROL}$ cells were passaged in culture and allowed to undergo telomere attrition-induced crisis. Clonal populations of NHA$_{TERT}$ cells emerged via endogenous reactivation of TERT expression[35]. NHA$_{ALT}$ cells were generated by silencing ATRX expression in NHA$_{CONTROL}$ cells[13]. NHA$_{TERT+IDHmut-}$ cells were generated by transfecting p53/pRb-deficient NHAs that do not express IDHmut with TERT[32,35]. NHA$_{ATRX- IDHmut-}$ cells were generated by silencing ATRX expression in p53/pRb-deficient NHAs that do not express IDHmut[13]. Patient-derived LGOG models: BT54 cells were derived from a patient carrying a WHO grade III oligodendroglioma and are maintained as neurospheres in serum-free medium[44–46]. They do not form tumors in vivo[44] and were, therefore, used only for cell studies. The SF10417 model was isolated from a

recurrent *IDHmut* LGOG tumor expressing *TERT* and has been shown to be a faithful LGOG model[6,54]. SF10417 cells were maintained as monolayers on laminin-coated plates, which makes it difficult to grow the number of cells needed for a typical MRS experiment ($\sim2-8 \times 10^7$ cells). They do, however, readily form orthotopic tumor xenografts in vivo and were, therefore, used for in vivo imaging studies. *Patient-derived LGA models*: BT142 and MGG119 models were isolated from patients carrying *IDHmut*-positive *ATRX*-deficient LGA tumors[47,51]. Both models have previously been shown to use the ALT pathway as measured by c-circle formation and T-SCE[13]. For a generation of BT142 TERT+ and MGG119 TERT+ cells, Phoenix A cells were transfected with pWZL-blast-hTERT and retroviral constructs were introduced into MGG119 and BT142 cells. Following blasticidin (25 µg/ml) selection for 14 days, polyclonal populations of MGG119 TERT+ and BT142 TERT+ cells were isolated. All cell lines were routinely tested for mycoplasma contamination, authenticated by short tandem repeat finger-printing (cell line genetics), and assayed within 6 months of authentication.

**Quantitative RT-PCR**. Gene expression was measured by quantitative RT-PCR. Briefly, total RNA was extracted (RNeasy mini kit, Qiagen) and reverse transcribed (SuperScript II kit, Thermo Fisher) according to the manufacturer's instructions. Relative mRNA levels were quantified using an SYBR Green quantitative RT-PCR kit (Sigma) and normalized to β-actin. A complete list of primer sequences used in this study is included in Supplementary Table 2.

**Telomerase activity**. Telomerase activity was confirmed via the telomeric repeat amplification protocol (TRAP) assay, which is considered to be the gold standard for measurement of telomerase activity[80]. The TRAPeze® RT kit (Sigma) was used according to the manufacturer's instructions.

**RNA interference**. Gene expression was transiently silenced by independent transfection with two nonoverlapping siRNA sequences/pools (Dharmacon) for 72 h[46,81]. For silencing *TERT*, Accell SMARTpool siRNA sequences (set of 4: GUAAAAUACUGAAUAUAUG, UCGACGUCUUCCUACGCUU, CGUACA GGUUUCACGCAUG, GGCUGAUGAGUGUGUACGU) and siGENOME SMARTpool siRNA sequences (set of 4: GGUAUGCCGUGGUCCAGAA, CCACGUCUCUACCUUGACA, UCACGGAGACCACGUUUCA, GCGUGGU GAACUUGCGGAA) were used. For silencing *GLS1*, ON-TARGETplus siRNA (CCUGAAGCAGUUCGAAAUA) and siGENOME SMARTpool siRNA sequences (set of 4: AGACAUGGUUGGUAUAUUA, UGAAUAAGAUGGCUGGUAA, GGUGGUUUCUGCCCAAUUA, GAAUAACACUCCCAUGGAU) were used. For *NAMPT*, OnTARGETplus siRNA (CAAGCAAAGUUUAUUCCUA) and siGENOME SMARTpool siRNA sequences (set of 4: GGUAAGAAGUUUCCUG UUA, UAAGGAAGGUGAAAUAUGA, CAAAUUGGAUUGAGACUAU, UAAC UUAGAUGGUCUGGAA) were used for silencing.

**ATRX re-expression**. For generation of the BT142 ALT− model, *ATRX* expression was transiently restored in BT142 neurospheres by electroporation with the plasmid IF-GFP-ATRX (gift from Michael Dyer, Addgene #45444) as described[12]. *ATRX* expression was verified at 72 h by quantitative RT-PCR and immunoblotting. Loss of the ALT phenotype was confirmed by quantification of c-circle levels and T-SCE.

**c-Circle assay**. The c-circle assay is considered to be the gold standard for rapid, quantitative, and robust assessment of the ALT phenotype[37,38]. c-circle levels were measured by telomeric qPCR with and without amplification by φ29 polymerase[82]. Briefly, for every sample, 16 ng of genomic DNA was mixed with 0.2 µg/µl bovine serum albumin, 0.1% Tween, 4 µM dithiothreitol, 1 mM of each dNTP, 3.75U of φ29 DNA polymerase, and 1× φ29 buffer and incubated at 30 °C for 8 h followed by 20 min at 65 °C. The telomeric content of samples with and without φ29 polymerase was calculated by qPCR and normalized to the respective single-copy gene reference (36B4). The following primer sequences were used: telomeric DNA (forward primer: GGTTTTTGAGGGTGAGGGTGAGGGTGAGGGTGAGGGT; reverse primer: TCCCGACTATCCCTATCCCTATCCCTATCCCTATCCCTA) and 36B4 (forward primer: CAGCAAGTGGGAAGGTGTAATCC; reverse primer: CCATTCTATCATCAACGGGTACAA). The c-circle level was calculated by subtracting the normalized telomeric DNA content of the sample without φ29 polymerase from the normalized telomeric DNA content of the sample with φ29 polymerase. U2OS cells were used as positive control.

**T-SCE assay**. T-SCE was measured by CO-FISH[13]. Briefly, cells were incubated with bromodeoxyuridine and colcemid, fixed, and metaphase spreads prepared on glass slides. These spreads were incubated with RNase, stained with Hoechst 33258, and UV-treated before exonuclease III digestion. Following formaldehyde fixation, metaphase spreads were treated with pepsin, fixed, ethanol dehydrated, and incubated with the leading strand probe (TelC-Cy3-red, PNA Bio). Slides were then washed with a formamide-based solution and PBS, dehydrated, incubated with the lagging strand probe (TelG-FITC-green, PNA Bio), stained with DAPI, and mounted for microscopy. Metaphase spreads without PNA probes were examined as negative controls in which no fluorescence was expected. The

appearance of yellow foci resulting from the overlap of the signal from the leading and lagging strand probes was considered to be representative of T-SCE. GM487 cells were used as a positive control as described previously[13]. For quantification, the percentage of chromosomes with T-SCE in a group of 50 cells was calculated.

**Spectrophotometric assays and immunoblotting**. NAD+, NADH, NADP+, NADPH, α-KG, alanine aminotransferase, lactate dehydrogenase, GLS, and NAMPT activity were determined using commercial kits (Abcam). For immuno-blotting, cell lysates were separated by SDS-PAGE, transferred onto PVDF membranes, and probed for ATRX using an antibody that has been validated by knock-down (Abcam, catalog # ab188027). β-actin (Cell Signaling, catalog # 4970) was used as a loading control.

**¹H-MRS and PCA of cell extracts**. Metabolites were extracted from cells or tumor tissues by dual-phase extraction[45,46,83]. ¹H-MRS spectra (1D ZGPR sequence, 90° flip angle, 3 s relaxation delay, 256 acquisitions) were obtained using an 11.7 T spectrometer (Bruker) equipped with a Triple Resonance CryoProbe. Peak integrals were quantified using Mnova (Mestrelab), corrected for saturation, and normalized to cell number and an external reference (trimethylsilyl propionate). Unsupervised PCA and supervised PLSDA were performed using SIMCA (Umetrics). VIP scores were calculated using MetaboAnalystR3.0 and metabolites with scores >1.0 considered discriminating metabolites[39].

**Alanine uptake assay**. For measurement of alanine uptake, cells ($\sim1 \times 10^7$) were incubated with a medium containing 10 mM [1-¹³C]-alanine for 24 h. Media samples were collected at 0 and 24 h and proton-decoupled, thermally-polarized ¹³C-MRS spectra were acquired on a Bruker 11.7 T spectrometer (30° flip angle, 3 s relaxation delay, 1024 acquisitions). Peak integrals for [1-¹³C]-alanine were quantified using Mnova (Mestrelab), corrected for saturation effects, and normalized to cell number and to an external reference (trimethylsilyl propionate) of known concentration. Alanine uptake was calculated as the difference in normalized fmol (Δfmol/cell) between 0 and 24 h.

**Hyperpolarized ¹³C-MRS of live cells**. Hyperpolarized [1-¹³C]-alanine was prepared as previously described[42]. An aliquot corresponding to a final concentration of 32 mM was polarized for ~1.5 h, dissolved in isotonic buffer (100 mM sodium phosphate, 0.3 mM EDTA, pH 5), and added to an NMR tube containing a suspension of live cells[40]. Due to the alkaline nature of the hyperpolarized alanine preparation (resulting from the use of NaOH[42]), care was taken at the end of all experiments to verify that the final solution showed a pH ~7.5. ¹³C spectra were acquired on a 14 T Bruker spectrometer with a 13° flip angle and 3 s TR for 300 s[83]. Data analysis was performed using Mnova (Mestrelab). For kinetic analysis of product formation, the product integral was normalized to maximum substrate integral and cell number. In addition, the area under the curve (AUC) for the product was normalized to the AUC for substrate and cell number. Since lactate was the predominant product peak in TERT models, the hyperpolarized lactate/alanine ratio was calculated. With ALT models, the hyperpolarized pyruvate/alanine ratio was used since pyruvate was the predominant product peak.

**Orthotopic tumor xenografts**. Animal studies were conducted in accordance with UCSF Institutional Animal Care and Use Committee guidelines. NHA$_{TERT}$, NHA$_{ALT}$, SF10417, or BT142 cells ($3 \times 10^5$ cells/10 µl) were intracranially injected into athymic nude rats (male, rnu/rnu homozygous, 5–6-weeks old, Envigo Laboratories)[73]. Twenty-three rats were used ($n = 3$ each for NHA$_{TERT}$ for slab dynamic and EPSI studies; $n = 3$ each for NHA$_{ALT}$ for slab dynamic and EPSI studies; $n = 3$ for tumor-free controls for slab studies; $n = 3$ for SF10417 and $n = 5$ for BT142 for EPSI). No animals were excluded from statistical analysis and investigators were not blinded in the studies.

**Anatomic MRI**. T2-weighted images were acquired on a horizontal 3 T scanner (Bruker Biospin) equipped with a dual-tuned ¹H-¹³C volume coil using a spin-echo TurboRARE sequence (TE/TR = 64/3484 ms, FOV = 43 × 43 mm², 256 × 256, slice thickness = 1 mm, NA = 6). Tumor contours in each axial slice were drawn manually and volume determined as a sum of the areas multiplied by slice thickness.

**In vivo hyperpolarized ¹³C-MRS studies**. For slab studies, following injection of 2.2 ml of hyperpolarized [1-¹³C]-alanine (final concentration of 96 mM) via a tail-vein catheter over 12 s, dynamic ¹³C spectra were acquired from a 15 mm slab through the brain every 3 s using a flyback spectral-spatial RF pulse[53] with a 30° flip angle on pyruvate and lactate and 3° flip angle on alanine. For imaging, a 2D flyback spectral-spatial EPSI pulse[53] was used (echo time = 7.76 ms, duration of the excitation pulse = 0.52 ms, spatial resolution of 5.375 × 5.375 × 8 mm³, field of view = 8 × 8 over 43 × 43 mm², temporal resolution of 3 s, spectral resolution of 128 points over a spectral bandwidth of 20 ppm). Bruker Biospin Paravision 6 software was used for data collection. The ¹³C spectra for slab studies were analyzed using Mnova. EPSI data was analyzed using in-house Matlab codes available with this paper as Supplementary Software as well as at a GitHub

repository (https://github.com/ViswanathLab/EPSI). For each voxel at every time point, peak integrals were evaluated and the lactate/alanine ratio (since lactate was the predominant product peak for LGOG models), pyruvate/alanine ratio (since pyruvate was the main product peak for LGA models) and alanine SNR calculated. EPSI data were quantified in magnitude mode. Color heatmaps were produced by interpolating the data using a Lanczos-2 kernel and normalizing to noise, which was calculated as the standard deviation of the real part of the signal in a voxel outside of the brain.

**Patient samples.** Snap-frozen human tumor tissue, with no patient-identifying information, was obtained in compliance with the informed consent policy from the UCSF Brain Tumor Center Biorepository and Pathology Core. Sample use was approved by the Committee on Human Research at UCSF. Samples were diagnosed as LGOG with *IDHmut* and 1p19q codeletion, diffuse LGA, IDHmut, or gliosis from epilepsy patients by a board-certified neuropathologist (J.J.P.). *TERT* expression and telomerase activity were confirmed by quantitative RT-PCR and TRAP assay respectively as described above. ALT status was investigated by measuring *ATRX* loss via quantitative RT-PCR and via the c-circle assay as described above. Eight samples each of LGOG, LGA, and gliosis biopsies were examined.

**Statistical analysis.** All experiments were performed independently on a minimum of three biological replicates ($n \geq 3$) and results presented as mean ± standard deviation. Statistical significance was assessed using an unpaired Student's $t$ test assuming unequal variance with $p < 0.05$ considered significant. Where applicable, correction for multiple comparisons was performed using the Holm–Šídák method. For all figures except Fig. 9e, * represents $p < 0.05$, ** represents $p < 0.01$, and *** represents $p < 0.005$. For the sake of clarity, statistical significance with $p < 0.005$ was represented as # for Fig. 9e. Data were analyzed using Microsoft Excel version 16 or GraphPad Prism 9.

**Reporting summary.** Further information on research design is available in the Nature Research Reporting Summary linked to this article.

## Data availability
Source data underlying the figures in this study are available as a source data file. All other data supporting the findings of this study are available within the paper, its Supplementary Information files, and from the corresponding author (P.V.) upon reasonable request. Cell lines used in this study are also available from the corresponding author (P.V.) upon reasonable request. Source data are provided with this paper.

## Code availability
The custom Matlab codes used for imaging data analysis in this study are available with the manuscript as Supplementary Software and at a GitHub repository (https://github.com/ViswanathLab/EPSI).

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

## Acknowledgements

The authors thank Anny Shai (UCSF Brain Tumor Center) for her help with the procurement and analysis of patient biopsies. We also thank Peng Cao (Larson Lab, UCSF Department of Radiology and Biomedical Imaging) for his help with EPSI design. This work was supported by the following grants: NIH R01CA239288, Department of Defense W81XWH201055315, UCSF Brain Tumor Center SPORE Career Enhancement Program Award (NIH P50CA97257), NIH P01CA118816, NICO, NIH R01CA197254, NIH R01CA172845, NIH R01NS105087, NIH P41EB013598, American Cancer Society Research Scholar Grant (131715-RSG-18-005-01-CCE), UCSF Brain Tumor Loglio Collective and NICO.

## Author contributions

P.V. and S.M.R. conceived and jointly supervised the research; P.V. designed and performed experiments and planned the studies. G.B. designed and performed in vivo imaging experiments. J.M. generated cell lines and performed experiments to verify ALT status. A.M.G. assisted with cell culture and in vivo studies. P.E.Z.L. provided intellectual contributions to the design of imaging pulse sequences. J.J.P. provided the patient biopsy samples. H.A.L. and J.F.C. provided cell lines. R.O.P. provided cell lines, reagents, and analytical tools. P.V. wrote the original paper. P.V. revised the paper. G.B., H.A.L., R.O.P., and S.M.R. reviewed and commented on the paper. P.V. and S.M.R. secured funding for the paper.

## Competing interests

The authors declare no competing interests.
