## [Peer Review File · Nature Communications]

REVIEWER COMMENTS

Reviewer #1 (Remarks to the Author); expert on glioma:

I read with great interest the manuscript entitled "Non-invasive assessment of telomere maintenance mechanisms in brain tumors".

The authors provide a unique set of approaches to study how ¹H-magnetic resonance spectroscopy (MRS) imaging can be used to study and monitor telomere maintenance in model systems of low-grade oligodendrogliomas (LGOGs) and low-grade astrocytomas (LGA). LGOGs activate the TERT pathway, while LGAs drive the alternative lengthening pathway (ALT). They first demonstrate that ALT and TERT impact the levels of metabolites in a distinct manner. Using this information, they utilized hyperpolarized alanine to distinguish TMM status in model systems of low grade gliomas. They found that ALT NHA (immortalized astrocytes) cells produce mainly pyruvate from alanine, whereas TERT NHA build up lactic acid. They went on to rescue these observations by silencing either TERT or ALT. They confirmed these findings in vivo as well. Using 2D EPSI of hyperpolarized [1-¹³C]-alanine they demonstrated in vivo that this technique can be used to distinguish between benign and malignant tissue as well. Finally, they linked the mechanisms to NAMPT and GLS to account for the differences in metabolization.

The strengths of the manuscript are the model systems and the translational imaging techniques. The weaknesses are the relative paucity with regards to highly detailed molecular mechanisms.

I have only a couple of remarks related to the scientific rigor of the studies.

1. They should use at least two oligonucleotides for siRNA experiments.
2. They should demonstrate the distribution of values, using dot-blot presentations of their diagrams.
3. They should provide the exact number of samples for each experiment. Are the samples biological or technical replicates?

Reviewer #2 (Remarks to the Author); expert on MRS and ¹³C-metabolism:

The manuscript describes a series of in vitro and in vivo experiments investigating the link between metabolic reprogramming and telomere maintenance mechanisms (TMMs) in glioma. Leveraging these associated metabolic changes, the authors used both ¹H MRS and MRS of hyperpolarized (HP) ¹³C-alanine to differentiate brain tumors and normal brain based on their TMM status. The translation of this approach could be beneficial for both diagnosis of brain tumors as well as treatment selection and response monitoring. The study is well designed and addresses an important topic with high potential for clinical impact for the patients with brain tumors. The manuscript is detailed and clear and it merits publication in Nature Communications. A few minor points should be addressed in a minor revision.

1. The authors found no difference in alpha-KG between NHA_{control} and NHA_{TERT} cells (Fig. 1G). Shouldn't then the sum of HP pyruvate and lactate be approximately the same for these cells (Fig. 2)? This does not seem to be the case. Also in Fig. 2D, it appears that the pyruvate peak has a local minimum at ~100 s before increasing again and then eventually decaying away. If true, how is this possible? The time course plot in Fig. 2E does not show this behavior.
2. The authors use the HP pyruvate/lactate ratio or its inverse as the metric to differentiate tumor from normal appearing brain tissue. Considering that "tumor-free healthy did not show metabolism of hyperpolarized [1-¹³C]-alanine to either [1-¹³C]-pyruvate or [1-¹³C]-lactate (Fig.S3A-S3B)" wouldn't a metric of pyruvate/alanine and/or lactate/alanine (or total ¹³C signal) be a more robust metric? Also, how is it possible that both pyruvate/lactate and lactate/pyruvate of normal brain are

less than 1 (Fig. S3D and E)? Similar question for contralateral brain EPSI data shown in Figs. 5 and 6. Also, it would be helpful if the lactate/alanine and pyruvate/alanine maps could also be shown for the EPSI data. The T2w MRI and heatmaps could be cropped to save space.

3. Judging from the pyruvate hydrate/pyruvate ratio the in vitro HP alanine measurements were done at higher than physiological pH (at least for the neurospheres, harder to say for the NHA cells). Is this correct and could this potentially have affected the results?

Minor comments:

1. Why was SF10417 chosen as the second in vivo model for TERT instead of BT54? The neurosphere experiments were done with BT54 and BT 142 cells.
2. Please add the duration of the excitation pulse, echo time, and either the imaging field of view or matrix size for the EPSI acquisition. Please also specify how the EPSI data were quantified, i.e., in magnitude or absorption mode.
3. Fig. 4 caption "1H-MRS and hyperpolarized [1-13C]-alanine metabolism can detect TMM status in vivo" and similar statements in the manuscript. However, the 1H-MRS measurements in these experiments were done on tissue extracts and not in vivo. Please rephrase.
4. p10: "There was a temporal shift between the maxima for alanine (..) and pyruvate (...) and a further shift between pyruvate and lactate (...), indicating that pyruvate and lactate production in NHA_{TERT} tumor-bearing animals originated from HP alanine metabolism within the tumor in the brain. " It is not clear how the spatial origin of the metabolite signals is related to the temporal shift. Please clarify.
5. On several occasions either pyruvate or lactate is referred to as the "predominant peak." This should be changed to product peak as alanine was certainly higher.
6. On two occasions "redox" is used as a noun. Please revise.
7. p7: "alanine was ... injected into live cells." I assume the alanine was added to the cells/cell solution?
8. Please define LGA and VIP. Also, AXP is used before its definition.
9. Ref. 48: Victor is the first name of the first author. Please correct the reference.

Reviewer #3 (Remarks to the Author); expert on telomere:

Within this manuscript the authors suggest that they can distinguish between ALT based and telomerase based telomere maintenance by non-invasive MRS-detectable metabolic signature analysis.

Since I am not an expert in this method, this review focuses on the telomere aspects of the manuscript.

Developing a non-invasive method to distinguish between ALT and TERT+ tumors would be extremely valuable, as it will determine the treatment course for a patient. The data presented here look highly promising to someone not familiar with the analysis technique.

However, it would be important to expand the panel of cell lines used. The authors used a single astrocyte line, which was rendered telomerase positive by hTERT expression and ALT positive by ATRX deletion in the background of IDH1 mutation. While these cells resemble some features of ALT, they are not truly ALT positive. Also, the authors fail to include only ATRX deleted or only IDH1 mutant cells in the analysis, which would be very important, considering that IDH1 is part of the metabolic pathways in question and ATRX is a chromatin modifier and therefore capable of changing multiple cellular pathways.

The authors should include other ALT and telomerase positive cell lines, preferentially isogenic ones. Such lines have been generated by the Reddel lab (RReddel@cmri.org.au) and the Decottignies lab (anabelle.decottignies@uclouvain.be), both known to generously share their material.

The authors use patient derived cell lines as well, which is very important. However, the TERT silencing shown in Fig S2C is not very efficient. Therefore the authors should demonstrate that

telomerase activity is indeed abolished in these cells, especially given the many observations that hTR is the limiting factor for telomere addition, not hTERT. Similarly re-expression of ATRX is not an optimal model to switch off the ALT mechanism, so the authors need to demonstrate that TSCE is indeed abolished in these cells.

Assaying tumor biopsies is undoubtedly the correct approach and lends much credibility to the author's approach. However, the optimal approach would be to do a double blind testing of tumor material, where TMM is determined by the author's approach, and then independently verified via telomerase assays and ALT marker analysis.

In summary, this is a highly promising approach, touches on an extremely important problem and has great potential. If the telomere aspects can be improved as suggested, this manuscript would be an outstanding contribution.

We thank the reviewers for their time, positive reviews and valuable feedback regarding our manuscript. We have now addressed their comments and revised the manuscript in accordance with their suggestions. Revisions made to the manuscript are highlighted in blue here and in the text.

REVIEWER 1

I read with great interest the manuscript entitled “Non-invasive assessment of telomere maintenance mechanisms in brain tumors”. The authors provide a unique set of approaches to study how ¹H-magnetic resonance spectroscopy (MRS) imaging can be used to study and monitor telomere maintenance in model system of low-grade oligodendrogliomas (LGOGs) and low-grade astrocytomas (LGA). LGOGs activate the TERT pathway, while LGAs drive the alternative lengthening pathway (ALT). They first demonstrate that ALT and TERT impact the levels of metabolites in a distinct manner. Using this information, they utilized hyperpolarized alanine to distinguish TMM status in model systems of low grade gliomas. They found that ALT NHA (immortalized astrocytes) cells produce mainly pyruvate from alanine, whereas TERT NHA build up lactic acid. They went on to rescue these observations by silencing either TERT or ALT. They

confirmed these findings in vivo as well. Using 2D EPSI of hyperpolarized [1-¹³C]-alanine they demonstrated in vivo that this technique can be used to distinguish between benign and malignant tissue as well. Finally, they linked the mechanisms to NAMPT and GLS to account for the differences in metabolization. The strengths of the manuscript are the model systems and the translational imaging techniques. The weaknesses are the relative paucity with regards to highly

detailed molecular mechanisms. I have only a couple of remarks related to the scientific rigor of the studies.

- 1) They should use at least two oligonucleotides for siRNA experiments.

We have now provided data with two, independent non-overlapping siRNA oligonucleotides for every silencing experiment (please refer to pages 9, 16 and 17 in the revised manuscript). The sequence information for the siRNA oligonucleotides is included in the methods section (see pages 29-30).

- 2) They should demonstrate the distribution of values, using dot-blot presentations of their diagrams.

Throughout the revised manuscript, results are now presented as scatter plots with all the data points shown.

- 3) They should provide the exact number of samples for each experiment. Are the samples biological or technical replicates?

All the samples are biological replicates. We have now clarified this point in the methods section (please refer to page 35 in the revised manuscript). The details of sample number are now included in the legend for each figure.

REVIEWER 2

The manuscript describes a series of in vitro and in vivo experiments investigating the link between

metabolic reprogramming and telomere maintenance mechanisms (TMMs) in glioma. Leveraging these associated metabolic changes, the authors used both ^1H MRS and MRS of hyperpolarized (HP) ^{13}C alanine to differentiate brain tumors and normal brain based on their TMM status. The translation of this approach could be beneficial for both diagnosis of brain tumors as well as treatment selection and response monitoring. The study is well designed and addresses an important topic with high potential for clinical impact for the patients with brain tumors. The manuscript is detailed and clear and it merits publication in Nature Communications. A few minor points should be addressed in a minor revision.

- 1) The authors found no difference in alpha-KG between $\text{NHA}_{\text{control}}$ and NHA_{TERT} cells (Fig. 1G). Shouldn't then the sum of HP pyruvate and lactate be approximately the same for these cells (Fig. 2)? This does not seem to be the case.

The reviewer is correct in asserting that we might expect the sum of hyperpolarized pyruvate and lactate to be similar between $\text{NHA}_{\text{CONTROL}}$ and NHA_{TERT} cells given the lack of difference in α -KG between these cells. To address this issue, we investigated if any of the other factors that control hyperpolarized alanine metabolism differed between these models (please refer to schematic in Fig. 2a). We now provide data to show that, indeed, expression of the alanine transporters ASCT2 and LAT2 is higher in NHA_{TERT} cells (and NHA_{ALT} cells) relative to $\text{NHA}_{\text{CONTROL}}$ (please see Supplementary Fig. 7a and results in pages 17-18). Importantly, higher transporter expression is associated with higher cellular uptake of $[1-^{13}\text{C}]$ -alanine from the medium (Supplementary Fig. 7b and results in page 17-18), potentially contributing to higher production of pyruvate and lactate in NHA_{TERT} and NHA_{ALT} cells relative to $\text{NHA}_{\text{CONTROL}}$. Furthermore, TERT expression is linked to higher ASCT2 and

LAT2 expression and elevated alanine import in the patient-derived BT54 LGOG model (please refer to Fig. 8g-8h and results in page 18). Similarly, the ALT pathway is linked to elevated ASCT and LAT2 expression and higher alanine import in the BT142 LGA model (Fig. 8i-8j and results in page 18). We have also extended these observations to LGOG and LGA patient biopsies (Fig. 9i and results in page 19). Collectively, our studies mechanistically link TERT expression and the ALT pathway to elevated alanine transporter expression and higher alanine import in LGOGs and LGAs.

These results are important in light of the observation that transporter expression is crucial for hyperpolarized ^{13}C imaging as evidenced by a recent study showing that expression of the monocarboxylate transporters MCT1 and MCT4 can be rate-limiting for the conversion of hyperpolarized $[1-^{13}\text{C}]$ -pyruvate to lactate¹. Previous studies have shown that MCT1 and MCT4 are silenced in low-grade gliomas^{2,3}, thereby potentially limiting the utility of hyperpolarized $[1-^{13}\text{C}]$ -pyruvate for low-grade glioma imaging⁴. In this context, the results presented in this manuscript showing higher ASCT2 and LAT2 expression and higher alanine import in LGOGs and LGAs highlight the potential utility of hyperpolarized $[1-^{13}\text{C}]$ -alanine for imaging these tumors. We have now modified the discussion to include this information (please refer to page 24).

Also in Fig. 2D, it appears that the pyruvate peak has a local minimum at ~ 100 s before increasing again and then eventually decaying away. If true, how is this possible? The time course plot in Fig. 2E does not show this behavior.

In the revised manuscript, we now provide a zoomed-in version of the spectral array of hyperpolarized $[1-^{13}\text{C}]$ -alanine metabolism in NHA_{TERT} cells and show that there are no

major variations in the pyruvate peak in NHA_{TERT} cells (please see Fig. 2d). Furthermore, presumably as a result of the very low levels of pyruvate in NHA_{TERT} cells, there are small (~3-12s) variations between biological replicates in the timepoint of the pyruvate maximum in these cells that are smoothed when the data is averaged for the time course in Fig. 2e (please refer to the source data file for Fig. 2e).

- 2) The authors use the HP pyruvate/lactate ratio or its inverse as the metric to differentiate tumor from normal appearing brain tissue. Considering that “tumor-free healthy did not show metabolism of hyperpolarized [1-13C]-alanine to either [1-13C]-pyruvate or [1-13C]-lactate (Fig.S3A-S3B)” wouldn’t a metric of pyruvate/alanine and/or lactate/alanine (or total 13C signal) be a more robust metric?

We agree with the reviewer that the hyperpolarized pyruvate/alanine ratio (for ALT cells) or the hyperpolarized lactate/alanine ratio (for TERT cells) is the better metric and have, therefore, revised the manuscript accordingly (please see Fig. 2g-2h, Fig. 3c, Fig. 3f, Fig. 4d, Fig. 4f, Fig. 6b, Fig. 6f, Fig. 7b, Fig. 7f, Supplementary Fig. 5d-5e, Supplementary Fig. 6e, Supplementary Fig. 6k; please also refer to associated text on pages 9-10, 12-16 and 33-34). In addition, the data for alanine, pyruvate and lactate levels for all our datasets have been presented in the source data file.

Also, how is it possible that both pyruvate/lactate and lactate/pyruvate of normal brain are less than 1 (Fig.S3D and E)? Similar question for contralateral brain EPSI data shown in Figs. 5 and 6.

The normal brain has very low levels of both pyruvate and lactate, resulting in pyruvate/lactate and lactate/pyruvate ratios ~ 1 . This is consistent across normal brain from tumor-free healthy animals (please refer to source data in source data file for revised Supplementary Fig. 5d-5e) and the contralateral brain from tumor-bearing LGOG (see source data for revised Fig. 6b and Fig. 6f) and LGA (refer to source data for Fig. 7b and Fig. 7f) models. Importantly, as suggested by the reviewer in the comment above, we have now replaced the pyruvate/lactate and lactate/pyruvate ratios with the pyruvate/alanine or the lactate/alanine ratios respectively.

Also, it would be helpful if the lactate/alanine and pyruvate/alanine maps could also be shown for the EPSI data. The T2w MRI and heatmaps could be cropped to save space.

In line with the reviewer's comment here and above, we have now revised the figures to show the heatmaps for lactate/alanine and pyruvate/alanine (please see revised Fig. 6d, Fig. 6h, Fig. 7d and Fig. 7h).

- 3) Judging from the pyruvate hydrate/pyruvate ratio the in vitro HP alanine measurements were done at higher than physiological pH (at least for the neurospheres, harder to say for the NHA cells). Is this correct and could this potentially have affected the results?

Due to the alkaline nature of the hyperpolarized [1-¹³C]-alanine preparation, we were careful in monitoring the pH during every experiment and consistently observed pH values ~ 7.5 . We, therefore, believe that our results have not been influenced by variations in pH. We have now modified the methods section to reflect this information (see page 33).

Minor comments:

- 1) Why was SF10417 chosen as the second *in vivo* model for TERT instead of BT54? The neurosphere experiments were done with BT54 and BT142 cells.

The BT54 cells readily grow as neurospheres in culture, but do not form tumors *in vivo*. The SF10417 cells grow as monolayers on laminin-coated plates, which makes it difficult to grow the number of cells needed for a typical MRS experiment ($\sim 2-8 \times 10^7$ cells). They do, however, readily form orthotopic tumor xenografts *in vivo*. Therefore, we used the BT54 cells for *in vitro* experiments and the SF10417 cells for *in vivo* imaging studies. We have now modified the methods section to reflect this information (please see pages 27-28).

- 2) Please add the duration of the excitation pulse, echo time, and either the imaging field of view or matrix size for the EPSI acquisition. Please also specify how the EPSI data were quantified, i.e., in magnitude or absorption mode.

We have now added this information to the methods section of the manuscript (please see page 34).

- 3) Fig. 4 caption “ ^1H -MRS and hyperpolarized [^{13}C]-alanine metabolism can detect TMM status *in vivo*” and similar statements in the manuscript. However, the ^1H -MRS measurements in these experiments were done on tissue extracts and not *in vivo*. Please rephrase.

We have now revised the manuscript in accordance with the reviewer’s suggestion (please refer to results on page 12 and the caption for revised Fig. 5a-5b on page 39).

- 4) p10: “There was a temporal shift between the maxima for alanine (..) and pyruvate (...) and a further shift between pyruvate and lactate (...), indicating that pyruvate and lactate production in NHATERT tumor-bearing animals originated from HP alanine metabolism within the tumor in the brain”. It is not clear how the spatial origin of the metabolite signals is related to the temporal shift. Please clarify.

This statement was based on the logic that if metabolism of hyperpolarized alanine was occurring in the liver or vasculature and alanine, pyruvate and lactate were being imported into the brain, we would potentially expect all 3 species to appear in the spectral array at the same time-point. The presence of a temporal shift between alanine, pyruvate and lactate, taken together with the lack of alanine metabolism to pyruvate or lactate in tumor-free healthy animals, suggests that the observed metabolism is likely to be originating in the tumor in the brain. We have now modified the results section to reflect this information (please refer to pages 13-14).

- 5) On several occasions either pyruvate or lactate is referred to as the “predominant peak.” This should be changed to product peak as alanine was certainly higher.

We have now revised the manuscript in accordance with the reviewer’s suggestion (please refer to pages 13-15 and pages 33-34) .

- 6) On two occasions “redox” is used as a noun. Please revise.

We have now revised the manuscript in accordance with the reviewer’s suggestion (please see pages 4 and 7).

7) p7: “alanine was ... injected into live cells.” I assume the alanine was added to the cells/cell solution?

The alanine was, indeed, added to a suspension of live cells. We have now clarified the text accordingly (please see page 8 in the revised manuscript).

8) Please define LGA and VIP. Also, AXP is used before its definition.

We have now revised the manuscript in accordance with the reviewer’s suggestion (please refer to pages 3, 5 and 6).

9) Ref. 48: Victor is the first name of the first author. Please correct the reference.

We apologize for this oversight and have now revised the manuscript in accordance with the reviewer’s suggestion (please see page 23).

REVIEWER 3

Within this manuscript the authors suggest that they can distinguish between ALT based and telomerase based telomere maintenance by non-invasive MRS-detectable metabolic signature analysis. Since I am not an expert in this method, this review focuses on the telomere aspects of the manuscript. Developing a non-invasive method to distinguish between ALT and TERT+ tumors would be extremely valuable, as it will determine the treatment course for a patient. The data presented here look highly promising to someone not familiar with the analysis technique.

1) However, it would be important to expand the panel of cell lines used. The authors used a single astrocyte line, which was rendered telomerase positive by hTERT expression and ALT

positive by ATRX deletion in the background of IDH1 mutation. While these cells resemble some features of ALT, they are not truly ALT positive.

In accordance with the reviewer's suggestion, we have now revised the manuscript to include information on the verification of the TMM status of all cell models used in this study, including the genetically engineered NHA models (please refer to Supplementary Table 1 for a summary and Supplementary Fig. 1 for the data). Specifically, we have confirmed the presence of the ALT pathway in NHA_{ALT} cells by measuring c-circles (please refer to Supplementary Fig. 1e), which are extratelomeric circles of DNA that are considered to be a specific, quantifiable hallmark of the ALT phenotype^{5,6}. The occurrence of telomeric sister chromatid exchange (T-SCE), another phenotypic characteristic of ALT cells, has previously been confirmed in NHA_{ALT} cells⁷. Furthermore, in line with previous results⁸, we have confirmed TERT expression as well as telomerase activity in NHA_{TERT} cells via quantitative RT-PCR and the telomeric repeat amplification protocol (TRAP assay) respectively (see Supplementary Fig. 1a-1b). We have revised the results section to reflect this information (please see pages 5-6).

Importantly, in line with the reviewer's suggestion (here as well as comment #3, see below), we have expanded the panel of cell lines analyzed in this manuscript. Previous studies suggest that exogenous TERT expression in ALT+ cells can potentially lead to a TERT+ phenotype, thereby providing an isogenic platform to interrogate TMM status⁹⁻¹¹. We now include data generated with 2 additional pairs of patient-derived models: 1) the MGG119 model, which has been shown to use the ALT pathway⁷, and an isogenic MGG119 model that has been engineered to express TERT and 2) the ALT-positive BT142 model⁷ as well as an isogenic BT142 line engineered to express TERT. We have verified the ALT and TERT

phenotypes of these models via measurement of c-circles (Supplementary Fig. 1e), T-SCE (Supplementary Fig. 1f), quantification of TERT mRNA (Supplementary Fig. 1a) and telomerase activity (Supplementary Fig. 1b). We show that our ^1H -MRS biomarkers as well as hyperpolarized alanine metabolism are in line with the data in the $\text{NHA}_{\text{CONTROL}}$, NHA_{TERT} , NHA_{ALT} , BT54 TERT+ and BT54 TERT-, BT142 ALT+ and BT142 ALT- and SF10417 models (please refer to Fig. 4a-4f and pages 10-12 of the results section). We have also modified the discussion to reflect the implications of this information for the non-invasive monitoring of the development of resistance to TERT and ALT inhibitors (please see page 25). Taken together, the vast number of cell lines analyzed in this study serves to emphasize the robustness of our metabolic data and imaging biomarkers.

- 2) Also, the authors fail to include only ATRX deleted or only IDH1 mutant cells in the analysis, which would be very important, considering that IDH1 is part of the metabolic pathways in question and ATRX is a chromatin modifier and therefore capable of changing multiple cellular pathways.

We agree with the reviewer that this is an important issue and have now revised the manuscript to include this information (please see pages 7-8 and Supplementary Fig. 3). Our ^1H -MRS studies identified similar levels of 2-hydroxyglutarate (2-HG), the product of IDHmut, in $\text{NHA}_{\text{CONTROL}}$, NHA_{TERT} and NHA_{ALT} cells (see Supplementary Fig. 3), thereby suggesting that the metabolic biomarkers identified in our study are linked to TERT expression or the ALT pathway, as opposed to IDHmut. Nevertheless, we have now examined the contributions of the IDH1 mutation (IDHmut), TERT and ATRX to the metabolic alterations linked to TERT expression or the ALT pathway in our studies.

Specifically, we have examined cells that express TERT or are ATRX-deficient in the absence of IDHmut (NHA_{TERT+IDHmut-} and NHA_{ATRX-IDHmut-} cells respectively) and compared to cells expressing only IDHmut (absence of TERT expression and presence of ATRX; NHA_{CONTROL}), cells expressing IDHmut and TERT (NHA_{TERT}) and cells expressing IDHmut that are ATRX-deficient (NHA_{ALT}; please refer to Supplementary Table 1 for a summary of the characteristics of the NHA lines).

Previous studies indicate that ATRX loss leads to activation of the ALT pathway only in the presence of IDHmut⁷. ATRX deletion in the absence of IDHmut does not lead to development of the ALT pathway⁷. The NHA_{ATRX-IDHmut-} cells, therefore, do not develop the ALT pathway, as evidenced by the lack of c-circles (previously in ref⁷ and in Supplementary Fig. 1e) and the lack of T-SCE (previously in ref⁷). Importantly, our results indicate that levels of glutamate, α -KG, alanine and AXP were elevated only in NHA_{ALT} cells relative to NHA_{CONTROL} and no alterations were observed in NHA_{ATRX-IDHmut-} cells (Supplementary Fig. 3), thereby linking these metabolites to the ALT pathway. Similarly, levels of TERT-linked metabolites i.e. GSH, NAD(P)/H, aspartate or AXP were not altered in NHA_{TERT+IDHmut-} cells and were elevated only in NHA_{TERT} cells relative to NHA_{CONTROL}. Collectively, these results link our metabolic biomarkers to TERT expression or the ALT pathway in the context of IDHmut. We have now modified the discussion to reflect this information (please see pages 22 and 26).

- 3) The authors should include other ALT and telomerase positive cell lines, preferentially isogenic ones.

As described in detail in response to comment #1 above, we have now included data from additional isogenic patient-derived models (MGG119 and BT142) that are known to use the ALT pathway⁷ and have now been engineered to express TERT. We validated the TERT and ALT phenotypes of these models (see Supplementary Table 1 for a summary and Supplementary Fig. 1 for the results). Our ¹H-MRS studies indicated that levels of TERT-linked metabolites including GSH, NAD(P)/H and aspartate were higher in MGG119 TERT+ and BT142 TERT+ neurospheres relative to their isogenic ALT+ counterparts, while levels of ALT-linked metabolites including glutamate, α -KG and alanine were reduced. AXP, which does not differentiate between TERT and ALT models, did not differ between the isogenic TERT+ and ALT+ neurospheres. Importantly, lactate production from hyperpolarized [1-¹³C]-alanine was higher in MGG119 TERT+ and BT142 TERT+ neurospheres relative to MGG119 ALT+ and BT142 ALT+ neurospheres respectively, consistent with data from the other genetically engineered and patient-derived models described in this manuscript. These results are described in detail in Fig. 4a-4f and pages 10-12 of the results section. Collectively, our results emphasize the ability of our metabolic imaging biomarkers to monitor modulation of TERT expression or the ALT pathway in isogenic, clinically-relevant, patient-derived glioma models (see also the discussion on page 25).

- 4) The authors use patient derived cell lines as well, which is very important. However, the TERT silencing shown in Fig S2C is not very efficient. Therefore, the authors should demonstrate that telomerase activity is indeed abolished in these cells, especially given the many observations that hTR is the limiting factor for telomere addition, not hTERT.

Similarly, re-expression of ATRX is not an optimal model to switch off the ALT mechanism, so the authors need to demonstrate that TSCE is indeed abolished in these cells.

We agree that this is an important point. We now provide data to show that telomerase activity as determined by the TRAP assay, is significantly reduced upon TERT silencing (please see Supplementary Fig. 1b and associated text on page 9). As requested by the reviewer, we also show that T-SCE is lost upon ATRX re-expression in the BT142 model (please see Supplementary Fig. 1g). In addition, throughout the manuscript, we have verified the TERT status (via quantitative RT-PCR for TERT expression and the TRAP assay for telomerase activity) or ALT status (via quantification of c-circles, assessment of T-SCE and quantitative RT-PCR for ATRX expression) of our models. Please see a summary of our cell models and their characteristics in Supplementary Table 1 and data in Supplementary Fig. 1.

- 5) Assaying tumor biopsies is undoubtedly the correct approach and lends much credibility to the author's approach. However, the optimal approach would be to do a double-blind testing of tumor material, where TMM is determined by the author's approach, and then independently verified via telomerase assays and ALT marker analysis.

We agree with the reviewer regarding the value of double-blind testing of patient biopsies. However, the assays for verification of TERT or ALT status (such as the TRAP assay and c-circle quantification) as well as the MRS experiments require experienced personnel with very specialized expertise. As a result of the COVID-19 pandemic, we are currently operating under reduced densities and have a shortage of such trained personnel. Nevertheless, we recognize the importance of data from patient biopsies and have now strengthened our manuscript by 1) increasing the number of LGOG, LGA and gliosis

biopsies (n = 8 each as opposed to n = 3) and 2) verifying TERT status in LGOG biopsies via the TRAP assay for telomerase activity in addition to quantitative RT-PCR for TERT expression (please see Fig. 9a-9b and page 19) and 3) verifying ALT status in LGA biopsies via quantification of c-circles in addition to quantitative RT-PCR for ATRX (please refer to Fig. 9c-9d and page 19).

- 6) In summary, this is a highly promising approach, touches on an extremely important problem and has great potential. If the telomere aspects can be improved as suggested, this manuscript would be an outstanding contribution.

We thank the reviewer for these positive comments and hope we have now sufficiently addressed the comments regarding the telomere-related aspects of this manuscript.

References

- 1 Rao, Y. *et al.* Hyperpolarized [1-(13)C]pyruvate-to-[1-(13)C]lactate conversion is rate-limited by monocarboxylate transporter-1 in the plasma membrane. *Proc Natl Acad Sci U S A* **117**, 22378-22389, doi:10.1073/pnas.2003537117 (2020).
- 2 Chesnelong, C. *et al.* Lactate dehydrogenase A silencing in IDH mutant gliomas. *Neuro Oncol* **16**, 686-695 (2014).
- 3 Viswanath, P. *et al.* Mutant IDH1 expression is associated with down-regulation of monocarboxylate transporters. *Oncotarget* **7**, 34942-34955 (2016).
- 4 Chaumeil, M. M. *et al.* Hyperpolarized (13)C MR imaging detects no lactate production in mutant IDH1 gliomas: Implications for diagnosis and response monitoring. *Neuroimage Clin* **12**, 180-189 (2016).

- 5 Henson, J. D. *et al.* DNA C-circles are specific and quantifiable markers of alternative-lengthening-of-telomeres activity. *Nat Biotechnol* **27**, 1181-1185, doi:10.1038/nbt.1587 (2009).
- 6 Henson, J. D. *et al.* The C-Circle Assay for alternative-lengthening-of-telomeres activity. *Methods* **114**, 74-84, doi:10.1016/j.ymeth.2016.08.016 (2017).
- 7 Mukherjee, J. *et al.* Mutant IDH1 Cooperates with ATRX Loss to Drive the Alternative Lengthening of Telomere Phenotype in Glioma. *Cancer Res* **78**, 2966-2977, doi:10.1158/0008-5472.Can-17-2269 (2018).
- 8 Ohba, S. *et al.* Mutant IDH1 Expression Drives TERT Promoter Reactivation as Part of the Cellular Transformation Process. *Cancer Res* **76**, 6680-6689 (2016).
- 9 Perrem, K., Colgin, L. M., Neumann, A. A., Yeager, T. R. & Reddel, R. R. Coexistence of alternative lengthening of telomeres and telomerase in hTERT-transfected GM847 cells. *Mol Cell Biol* **21**, 3862-3875 (2001).
- 10 Grobelny, J. V., Kulp-McEliece, M. & Broccoli, D. Effects of reconstitution of telomerase activity on telomere maintenance by the alternative lengthening of telomeres (ALT) pathway. *Hum Mol Genet* **10**, 1953-1961, doi:10.1093/hmg/10.18.1953 (2001).
- 11 Perrem, K. *et al.* Repression of an alternative mechanism for lengthening of telomeres in somatic cell hybrids. *Oncogene* **18**, 3383-3390, doi:10.1038/sj.onc.1202752 (1999).

REVIEWERS' COMMENTS

Reviewer #1 (Remarks to the Author):

The authors have addressed my concerns. I feel that the paper has been improved through the revision. The studies are highly interesting to the brain tumor research community.

Reviewer #2 (Remarks to the Author):

The authors have more than adequately addressed the previously raised issues.

Reviewer #3 (Remarks to the Author):

I am satisfied with the revision and the additional data that was added. As referee focusing on the telomere aspects of the manuscript I am in support of publication and thank the authors for the in-depth address of my concerns.

We thank the reviewers for their thoughtful comments on our manuscript. Our response to the final comments from the reviewers is highlighted in blue below.

Reviewer #1 (Remarks to the Author):

The authors have addressed my concerns. I feel that the paper has been improved through the revision. The studies are highly interesting to the brain tumor research community.

Thank you for your positive feedback and encouraging review of our study.

Reviewer #2 (Remarks to the Author):

The authors have more than adequately addressed the previously raised issues.

Thank you for the constructive review of our manuscript.

Reviewer #3 (Remarks to the Author):

I am satisfied with the revision and the additional data that was added. As referee focusing on the telomere aspects of the manuscript I am in support of publication and thank the authors for the in-depth address of my concerns.

Thank you for the detailed review which has helped to improve the quality of our manuscript.